# Synthesis and Characterization of Novel Heterocyclic Chalcones from 1-Phenyl-1*H*-pyrazol-3-ol

**DOI:** 10.3390/molecules27123752

**Published:** 2022-06-10

**Authors:** Arminas Urbonavičius, Graziana Fortunato, Emilija Ambrazaitytė, Elena Plytninkienė, Aurimas Bieliauskas, Vaida Milišiūnaitė, Renzo Luisi, Eglė Arbačiauskienė, Sonata Krikštolaitytė, Algirdas Šačkus

**Affiliations:** 1Department of Organic Chemistry, Kaunas University of Technology, Radvilėnų Pl. 19, 50254 Kaunas, Lithuania; arminas.urbonavicius@ktu.lt (A.U.); grafortuna1996@gmail.com (G.F.); emilija.ambrazaityte@ktu.edu (E.A.); elena.plytninkiene@ktu.lt (E.P.); vaida.milisiunaite@ktu.lt (V.M.); sonata.krikstolaityte@ktu.lt (S.K.); 2Institute of Synthetic Chemistry, Kaunas University of Technology, K. Baršausko g. 59, 51423 Kaunas, Lithuania; aurimas.bieliauskas@ktu.lt; 3Department of Pharmacy—Drug Sciences, University of Bari “Aldo Moro”, Via E. Orabona 4, 70125 Bari, Italy; renzo.luisi@uniba.it

**Keywords:** Claisen–Schmidt condensation, heterocyclic chalcones, pyrazole, pyridine, 1,2-oxazole (isoxazole), ^15^N-labeled 1,2-oxazole

## Abstract

An efficient synthetic route to construct diverse pyrazole-based chalcones from 1-phenyl-1*H*-pyrazol-3-ols bearing a formyl or acetyl group on the C4 position of pyrazole ring, employing a base-catalysed Claisen–Schmidt condensation reaction, is described. Isomeric chalcones were further reacted with *N*-hydroxy-4-toluenesulfonamide and regioselective formation of 3,5-disubstituted 1,2-oxazoles was established. The novel pyrazole-chalcones and 1,2-oxazoles were characterized by an in-depth analysis of NMR spectral data, which were obtained through a combination of standard and advanced NMR spectroscopy techniques.

## 1. Introduction

Chalcones (or 1,3-diaryl-2-propen-1-ones) are widely distributed in naturally occurring products produced by bacteria, fungi, and numerous plant species. Chalcones do not accumulate in natural sources and serve as intermediates for flavanoid biosynthesis [1,2,3]. Chalcone-rich sources are highly valued, as they possess beneficial biological properties [1]. For example, licochalcone A demonstrated antibacterial effects against *B. subtilis*, human pathogenic *Mycobacteria* and *Legionella* species [4,5] and inhibited the growth of both *Leishmania major* and *Leshmania donovani* promastigotes and amastigotes or *P. falciparum* strains [6,7]. Isobavachalcone showed antifungal effects against *Candida albicans* and *Cryptococcus neoformans* [8]. Moreover, xanthohumol demonstrated antiviral properties against bovine viral diarrhoea virus, HSV-1 (herpes simplex virus) and HSV-2, CMV (cytomegalovirus) [9], and coronaviruses [10] and showed anti-HIV-1 activity [11]. Xanthohumol caused a dose-dependent decrease in the growth of human breast cancer (MCF7) [12,13], colon cancer (HT-29) [14,15], ovarian cancer (A-2780) [16] and prostate cancer cells in vitro [17]. 

The structural simplicity and therapeutic potential have motivated the design and development of synthetic chalcones with enhanced activity and potency [18]. 

Chalcones are usually synthesized from aromatic aldehydes and aliphatic aldehydes or ketones via the Claisen–Schmidt condensation reaction in the presence of base or acid catalysts [19,20,21,22]. Other procedures were efficiently employed for the synthesis of chalcones, including the Pd-catalysed Suzuki cross-coupling reaction between the appropriate cinnamoyl chloride and phenylboronic acid or benzoyl chloride and phenylvinylboronic acid or Heck coupling reaction between aryl iodide and an unsaturated ketone [23,24,25]. The Wittig olefination reaction of triphenylbenzoylmethylene phosphoranes and benzaldehydes and the Julia–Kocienski olefination technique of heteroaryl-sulfonyl phenylethanones and benzaldehydes were also applied to give chalcones in efficient yields [26].

Molecular modeling studies of chalcones using DFT methods indicated that 1,3-diaryl-2-propen-1-ones have two isomers, the (*E*)-isomer being thermodynamically more stable than the (*Z*)-isomer [27]. The (*E*)-chalcone derivatives are synthesized far more easily than the (*Z*)-isomer, and there have been only a few reports concerning the synthesis of the (*Z*)-isomers [28,29]. However, Yoshizawa et al. reported the synthesis of (*Z*)-chalcones from 1,3-diaryl-2-propynyl silyl ethers by a catalytic reaction using potassium *tert*-butoxide under acid treatment, in high yields and stereoselectivity [29]. Rajakumar et al. induced the photochemical *E*- to *Z*- isomerization of chalcone derivatives [30].

Among the synthetic chalcone derivatives, heterocyclic chalcones are important for medicinal chemistry, as most biologically active chemical entities contain a heterocyclic scaffold [31,32]. For example, Pd(II) or Pt(II) complexes containing chalcone **I** displayed good anticancer and antimicrobial activities [33], while the structure of pyridine-chalcone derivative **II** was developed as a potential anti-tubulin agent, with antiproliferative activity against a panel of cancer cell lines (Figure 1) [34]. The anticancer activity was also reported for indolizinyl compound **III**, with the potential to induce the caspase-dependent apoptosis of human lymphoma cells [35] or quinoxalinyl derivative **IV**, which was active against MCF-7-cell lines [36], and thiophen-2-yl derivative **V**, which was active against colorectal carcinoma cells by causing apoptosis [37]. Chalcone **VI** showed remarkable inhibition potency against AChE and MAO-B enzymes and, therefore, can be further developed as a novel, phenothiazine-based, dual-targeting inhibitor for neurogenerative diseases [38], while compound **VII** acts as a tissue transglutaminase inhibitor [39]. Compound **VIII** and analogues were designed as promising anti-tubercular agents by combining in silico design, QSAR-driven virtual screening, synthesis, and experimental evaluation. The synthesized nitroaromatic chalcone derivatives were also active against *Mycobacterium tuberculosis* strains resistant to isoniazid or rifampicin [40]. In addition, indole-based chalcone **IX** was reported to act as a nonselective COX-1 and COX-2 inhibitor and showed anti-inflammatory and antioxidant activities in vivo [41]. Moreover, pyrazole-based chalcone derivatives **X**–**XII** were also synthesized and investigated. Compound **X** was evaluated for its anti-inflammatory activity, and compounds **XI** and **XII** showed potential activity as chemotherapeutic agents for the treatment of hepatocellular carcinoma (HCC), as they caused cell cycle arrest at the G2/M phase and induced apoptotic cell death [42,43].

Among heterocycles, pyrazoles are considered privileged scaffolds in medicinal chemistry [32]. Pyrazole derivatives are known to exhibit anti-inflammatory, analgesic, anticancer, antimicrobial, anti-infective and other activities [44,45,46,47]. In recent publications, we have reported the synthesis and antimitotic activity of 2,4- or 2,6-disubstituted- and 2,4,6-trisubstituted-2*H*-pyrazolo[4,3-*c*]pyridines [48], the antiproliferative activity of 2,4,6,7-tetrasubstituted-2*H*-pyrazolo[4,3-*c*]pyridines [49], the photodynamic properties in the human skin melanoma cell line G361 of pyrazole-indole hybrids [50] and *N*-aryl-2,6-diphenyl-2*H*-pyrazolo[4,3-*c*]pyridin-7-amines [51] and the anthelmintic activity of benzopyrano[2,3-*c*]pyrazol-4(2*H*)-ones [52].

In continuation of our interest in the efficient synthesis of pyrazole-containing polycyclic systems starting from easily accessible 3-hydroxy-1-phenyl-1*H*-pyrazole, we report herein the synthesis and structural elucidation of novel, diverse pyrazole-chalcone derivatives via the base-catalysed, Claisen–Schmidt condensation reaction of 4-formyl or 4-acetyl-1-phenyl-1*H*-pyrazol-3-ols and appropriate acetophenones or carbaldehydes.

## 2. Results and Discussion

The synthesis of various (*E*)-3-(3-alkoxy-1-phenyl-1*H*-pyrazol-4-yl)-1-phenylprop-2-en-1-ones **4a**–**k** was carried out, as depicted in Figure 1. It started with the easily accessible 1-phenyl-1*H*-pyrazol-3-ol **1**, which was converted to 3-methoxy-, 3-propoxy-, 3-(2-methoxyethoxy)- and 3-benzyloxy-1-phenyl-1*H*-pyrazoles **2a**–**d** and corresponding 4-carbaldehydes **3a**–**d** via *O*-alkylation and Vilsmeier–Haack formylation procedures in a similar manner to what we described earlier [48,53,54,55,56]. The obtained compounds **3a**–**d** were then subjected to a Claisen–Schmidt condensation reaction with variously 4′-substituted acetophenones in the presence of ethanolic sodium hydroxide. The heating reaction mixture at 55 °C for 30 min afforded chalcones **4a**–**k** in fair to excellent yields (58–97%). A similar synthetic approach towards pyrazole-chalcones employing alcoholic NaOH-catalysed, Claisen–Schmidt condensation of 3-aryl-1*H*-pyrazole-4-carbaldehydes and acetophenones was also demonstrated by Aneja et al. in the course of pyrazolylpyrazolines [57] or by Baytas et al. for the preparation of 1,3-diarylpyrazoles [43].

An in-depth analysis of NMR spectral data, which were obtained through a combination of standard and advanced NMR spectroscopy techniques, such as ^1^H-^13^C HMBC, ^1^H-^13^C HSQC, ^1^H-^13^C H2BC, ^1^H-^15^N HMBC, ^1^H-^15^N LR-HSQMBC, ^1^H-^1^H TOCSY, ^1^H-^1^H COSY, ^1^H-^1^H NOESY and 1,1-ADEQUATE experiments, provided the key information in the establishment of structural assignments and predominant configuration, due to conformations in a solvent of novel pyrazole-chalcones. The synthesis and biological activity of the related compounds has been reported in previous works, but no data on conformational analysis supported by NMR experiments were given [58,59,60,61].

In the case of compound **4a**, the key information for structure elucidation was obtained from the ^1^H-^13^C HMBC, ^1^H-^13^C H2BC, ^1^H-^13^C HSQC and ^1^H-^15^N LR-HSQMBC spectral data (Figure 2). For instance, the pyrazole 5-H proton (singlet, δ 7.96 ppm) was easily distinguished, as it exhibited not only long-range HSQMBC correlations with neighboring N-1 “pyrrole-like” (δ −183.7 ppm) and N-2 “pyridine-like” (δ −118.7 ppm) nitrogen atoms, but also HMBC correlations with the quaternary carbons C-3 (δ 163.3 ppm) and C-4 (δ 107.0 ppm), respectively. The ^1^H-^13^C HSQC spectrum indicated that the aforementioned proton had a one-bond connectivity with carbon C-5 (δ 129.0 ppm), thus completing our assignment of the 1*H*-pyrazol-4-yl moiety. Moreover, these findings unambiguously confirmed the connectivity with the neighboring 1-phenylprop-2-en-1-one fragment via long-range HMBC correlations of the olefinic H_a_ proton and the aforementioned pyrazole ring carbons. The *E*-configuration at the C=C double bond unequivocally follows from the magnitude of the vicinal coupling between the olefinic protons H_a_ (δ 7.74 ppm) and H_b_ (δ 7.60 ppm), which exhibited an AB-spin system and appeared as two sets of doublets (^3^*J*_Ha,Hb_ = 15.5 Hz). As expected, the ^1^H-^13^C HMBC spectrum revealed distinct, long-range correlations between these olefinic protons and the phenyl group 2″(6″)-H protons (δ 8.02–8.03 ppm), with the characteristic signal of a carbonyl carbon (δ 190.7 ppm). The ^1^H-^1^H NOESY spectrum of **4a** further elucidated the connectivities based on through-space correlations. In this case, distinct NOEs were exhibited between the pyrazole ring proton 5-H and the olefinic proton H_a_, while the phenyl group 2″(6″)-H protons displayed correlation with the olefinic proton H_b_, thus allowing different structural fragments to be joined.

The obtained (*E*)-3-[3-(benzyloxy)-1-phenyl-1*H*-pyrazol-4-yl]-1-phenylprop-2-en-1-ones **4g**–**i** were further treated with trifluoroacetic acid for debenzylation. (*E*)-3-(3-Hydroxy-1-phenyl-1*H*-pyrazol-4-yl)-1-phenylprop-2-en-1-ones **5a**–**c** were obtained in fair to good yields (63–83%), as outlined in Figure 2. As expected, the cleavage of the OCH_3_ group (compound **5b**) under the given conditions was not observed.

Furthermore, we investigated the applicability of pyrazol-3-ol **1** as a starting material for the synthesis of (*E*)-1-(3-hydroxy-1-phenyl-1*H*-pyrazol-4-yl)-3-phenylprop-2-en-1-ones **8a**–**l** (Figure 3). Compound **1** was converted to pyrazol-3-yl acetate **6**, which was further subjected to Fries rearrangement reaction conditions, as previously described [48,56]. The obtained 4-acyl-3-hydroxy-1-phenyl-1*H*-pyrazole **7** was reacted with various (het)aromatic carbaldehydes under the Claisen–Schmidt reaction conditions to form targeted chalcones **8a**–**l**. Stirring the reaction mixture of pyrazole **7** and benzaldehyde in EtOH in the presence of NaOH at 55 °C [57] led to the formation of chalcone **8a** with a 88% yield. The same reaction conditions were applied to synthesize a series of novel (2*E*)-1-(3-hydroxy-1-phenyl-1*H*-pyrazol-4-yl)-3-phenylprop-2-en-1-ones **8b**–**l**. Most chalcones were obtained in fair to excellent yields (58–95%); when 4-(dimethylamino)benzaldehyde, pyridine-3-, pyridine-4- and thiophene-2-carbaldehydes were utilized for the condensation, lower yields of the appropriate products **8f,h,i**, and **k** were obtained (25–48%).

To obtain (*E*)-1-(3-alkoxy-1-phenyl-1*H*-pyrazol-4-yl)-3-(het)arylprop-2-en-1-ones **9a**–**i**, appropriate pyrazol-3-ols **8** were further *O*-alkylated using appropriate methyl-, propyl- and 2-methoxyethylhalides to produce alkoxy derivatives **9a**–**i** in fair to excellent yields (40–96%).

A representative 3-phenylprop-2-en-1-one **9a** showed distinct, long-range correlations in the ^1^H-^13^C HMBC, ^1^H-^13^C H2BC and ^1^H-^15^N HMBC spectra, which in combination with the data from the 1,1-ADEQUATE experiment, allowed us to provide unambiguous assignments of the ^1^H, ^13^C and ^15^N NMR resonances (Figure 3). For example, the pyrazole 5-H proton (singlet, δ 8.42 ppm) exhibited not only long-range HMBC correlations throughout the 1*H*-pyrazol-4-yl moiety, but also a strong correlation with the most downfield ^13^C resonance, which confidently was assigned to a carbonyl carbon (δ 183.3 ppm). This finding, in combination with the data from the 1,1-ADEQUATE experiment, allowed the adjacent protonated carbon to be assigned to the signal at δ 124.3 ppm, which showed a sole correlation with the aforementioned carbonyl carbon. Moreover, the protonated carbon also shared a correlation with an adjacent olefinic carbon that resonated at δ 142.6 ppm. With this information, the ^1^H-^13^C HSQC spectral data were applied to identify olefinic protons H_a_ (δ 7.63 ppm) and H_b_ (δ 7.82 ppm), which exhibited an AB-spin system and appeared as two sets of doublets (^3^*J*_Ha,Hb_ = 15.7 Hz). The ^1^H-^1^H NOESY spectrum of **9a** exhibited NOEs between the phenyl group 2″(6″)-H protons and both olefinic protons H_a_ and H_b_, while proton H_a_ also had an NOE with a pyrazole 5-H proton, which confirms their proximity in space.

To expand the structural diversity of the pyrazole-based, chalcone derivatives, we also employed Claisen–Schmidt reaction conditions for the synthesis of 1-(1-phenyl-1*H*-pyrazol-4-yl)-3-phenylprop-2-en-1-ones with pyridin-3-yl or pyridin-4-yl substituents on the third position of the pyrazole ring (Figure 4). We have previously demonstrated that 1-(3-hydroxy-1-phenyl-1*H*-pyrazol-4-yl)ethan-1-one can be efficiently converted to various 1-(3-aryl-1-phenyl-1*H*-pyrazol-4-yl)ethan-1-ones via *O*-triflation and Pd-catalysed Suzuki, Sonogashira or Heck reaction sequences [56]. In this research, 4-acetyl-1-phenyl-1*H*-pyrazol-3-yl trifluoromethanesulfonate **10** was subjected to a Pd-catalysed, Suzuki cross-coupling reaction with pyridin-3-yl and pyridin-4-yl boronic acids. When the cross-coupling reaction of triflate **10** and pyridin-3-yl boronic acid was performed under conventional heating in the presence of Pd(PPh_3_)_4_ and K_3_PO_4_ in refluxing dioxane, the starting materials decomposed. Refluxing the reaction mixture of coupling partners in EtOH in the presence of Pd(OAc)_2_ and Cs_2_CO_3_ led to the formation of product **11a** with only a 20% yield. The best result of Suzuki cross-coupling was accomplished using Pd(PPh_3_)_4_ as a catalyst and Cs_2_CO_3_ as a base and by performing the reaction at 80 °C with a microwave irradiation potency of 150 W in EtOH. 1-[1-Phenyl-3-(pyridinyl)-1*H*-pyrazol-4-yl]ethan-1-ones **11a,b** were obtained in 63–64% yields. The reaction was carried out in the presence of KBr, which is known to suppress triflate reduction by stabilizing the cationic (σ-aryl)-palladium transition state [62]. 4-Acetylpyrazoles **11a,b** were further employed in the Claisen–Schmidt condensation reaction with different carbaldehydes. The reaction was performed under the above-described conditions in the presence of ethanolic NaOH at 55 °C. As a result, when 1-[1-phenyl-3-pyridinyl-1*H*-pyrazol-4-yl]ethan-1-ones **11a,b** were reacted with benzaldehyde, 4-methyl- or 4-(trifluoromethoxy)benzaldehyde *E*- and *Z*-chalcones (**12a**–**f** and **13a**–**f**, respectively) were obtained in fair to good total yields (51–70%). The NMR spectra of inseparable mixtures showed the presence of both isomers in different ratios, with a predominance of the *E*-isomer. In contrast, compounds **12g**–**j** were obtained only as pure *E*-isomers. This latter observation can be explained by the strong electron-donor capacity of the 4-methyloxy group based on the resonance structure, which results in increased electron density of the enone moiety, while the electronegativity of the 4-trifluoromethyl group has a significance for decreased electron density of the enone moiety. It is known in most cases that the *E*-isomer is more stable from the perspective of thermodynamics, which makes it the predominant configuration among the chalcones [27].

It is widely accepted that a large and constant difference in the magnitudes of the ^3^*J*_HH_ coupling constants of the olefinic protons in *E*-*Z* isomers can be used for structural elucidation, which in our case were larger by approximately 3 Hz for the predominant *E*-isomer (15.6–15.7 Hz), while the minor *Z*-isomer provided significantly lower coupling constant values (12.7–12.8 Hz). Moreover, the 1D selective NOESY experimental data clearly showed that upon irradiation of the olefinic protons of the minor *Z*-isomer, the expected NOEs between them were observed. In the case of the major *E*-isomer, the olefinic protons exhibited only appropriate correlations with neighboring aromatic protons, therefore, unambiguously confirming the correct configuration (Appendix A).

As expected, the NMR spectral data of compound **12g** revealed a distinct difference in chemical shifts in the 1*H*-pyrazol-4-yl moiety compared with the other series of pyrazolo-chalcones, due to the pyridin-3-yl substituent on the third position of the pyrazole ring (Figure 4). The key information for structure elucidation of the pyridin-3-yl moiety was obtained from the ^1^H-^1^H TOCSY spectrum. The results clearly showed a spin system of four protons, which were mostly downfield. Moreover, a comparison between the ^1^H-^1^H COSY spectra and the ^1^H-^1^H TOCSY spectra showed a complete absence of COSY cross-peaks between one of the protons, with the remainder from the aforementioned spin system. This finding strongly hinted at a neighboring quaternary carbon at site 3′′′, which was unambiguously assigned from 1,1-ADEQUATE spectral data, where the protonated pyridine carbons C-2′′′ (δ 149.0 ppm) and C-4′′′ (δ 135.9 ppm) showed a sole correlation with C-3′′′ at δ 127.5 ppm. The remainder of the protonated pyridine carbons were easily assigned from the appropriate correlations in the ^1^H-^13^C H2BC spectrum. The 3-(pyridinyl)-1*H*-pyrazol-4-yl heterocyclic system contains three nitrogen atoms. The chemical shifts of the N-1 and N-2 atoms of compound **12g** were δ −161.8 and δ −78.2 ppm, respectively. The pyridin-3-yl substituent nitrogen resonated at δ −70.6 ppm.

Chalcones are versatile synthons for the synthesis of five- and six-membered nitrogen heterocycles, such as pyrazoles, pyrazolines, isoxazoles, isoxazolines, pyridines, pyrimidines, and others [63]. This was briefly demonstrated in this work by the treatment of compounds **4a** and **9a** with *N*-hydroxy-4-toluenesulfonamide (TsNHOH), in the presence of NaOH in EtOH/H_2_O (9:1 *v*/*v*) [64,65,66] (Figure 5). When a chalcone **4a** was used as a substrate, a regioselective formation of 3-(1*H*-pyrazol-4-yl)-5-phenyl-1,2-oxazole **14** was observed in 56% yield, while 5-(1*H*-pyrazol-4-yl)-3-phenyl-1,2-oxazole **15** was formed in a lower yield (36%), using chalcone **9a** as a starting material. The efforts to obtain pyrazole-isoxazoles from chalcones using more prevalent reaction conditions reported in the literature [67,68], i.e., treating compound **9a** with hydroxylamine in the presence of NaOH in MeOH/H_2_O (95/5 *v*/*v*), led to a mixture of 1,2-oxazoles **14** and **15** with a poor total yield of 23%. The formation of intermediate reaction products, such as isoxazolines or oximes, could be also identified by HPLC/MS data (HPLC data of crude reaction mixture are provided in Appendix A, followed by MS data in Appendix A).

The regioselective formation of pyrazole-isoxazoles **14** and **15** was confirmed by NMR studies (Figure 5). As expected, the ^1^H, ^13^C and ^15^N NMR chemical shifts and the relevant correlations in the two-dimensional NMR spectra of these two isomeric 1,2-oxazoles were highly similar. The unambiguous formation of 1,2-oxazole (isoxazole) moiety was easily deduced from ^1^H-^15^N HMBC spectral data, as it clearly showed a distinct long-range correlation between the isoxazole methine 4′-H proton and nitrogen N-2′, which resonated at δ −18.6 and −19.6 ppm for compounds **15** and **14**, respectively, and this is in good agreement with the data reported in the literature [69]. The 2 Hz optimized ^1^H-^15^N HMBC spectra hinted in favor of these structures. For instance, the conversion of chalcone **4a** provided an 1,2-oxazole derivative, in which the pyrazole 5-H proton (singlet, δ 8.33 ppm) exhibited not only long-range HMBC correlations throughout the 1*H*-pyrazol-4-yl moiety, but also a weak correlation with the oxazole N-2′ nitrogen at δ −19.6 ppm was observed. Meanwhile, the 1,2-oxazole derivative obtained from chalcone **9a** was assigned to structure **15**, due to the correlation with the neighboring protons 2″(6″)-H (δ 7.88 ppm) from the phenyl moiety. The aforementioned protons from the pyrazole and phenyl moieties in the ^1^H-^13^C HMBC spectrum showed three-bond connectivities with the appropriate isoxazole quaternary carbons C-3′ and C-5′, which allowed us to confirm the correct structure assignments afterwards via the analysis of *J*_CN_ couplings.

Then, in order to avoid any ambiguity in the structure assignment of regioisomeric 1,2-oxazoles, the ^15^N-labeled pyrazole-isoxazoles **16** and **17** were synthesized by analogy to **14** and **15**. The treatment of chalcone **9a** with ^15^N-hydroxylamine hydrochloride produced an inseparable mixture of regioisomers **16** and **17** in a ratio of about 8:1 (Figure 6). The selective ^15^N-labeling in azaheterocycles is an important method for studying molecular structures, which significantly expands the possibilities of using standard NMR methods [70]. The ^15^N-labeled aromatic heterocyclic structures typically have well-resolved ^1^H-^15^N (*J*_HN_) and ^13^C-^15^N (*J*_CN_) coupling constants, including additional splitting of the corresponding signals in the standard proton decoupled 1D ^13^C NMR and 1D ^1^H NMR spectra [71,72].

In the case of ^15^N-labeled pyrazole-isoxazoles **16** and **17**, the analysis of ^1^H-^15^N (*J*_HN_) coupling constants ^3^*J*_H4′-__N2′_ did not provide significant information regarding the correct structure confirmation, and were 1.23 Hz and 1.31 Hz for major and minor regioisomers, respectively. As expected, the unambiguous structure assignment of regioisomeric 1,2-oxazoles was achieved after a careful analysis of the ^13^C-^15^N (*J*_CN_) coupling constants, which were obtained from a ^13^C NMR spectrum. The ^13^C-^15^N spin–spin interaction was observed for the signals of the major regioisomer C-3′ (^1^*J*_C3′-N2′_ = 2.89 Hz), C-4′ (^2^*J*_C4′-N2′_ = 1.23 Hz) and C-5′ (^2^*J*_C5′-N2′_ = 1.39 Hz) from the 1,2-oxazole moiety, as well as the ^2^*J*_CN_ and ^3^*J*_CN_ couplings from the adjacent phenyl ring. The minor regioisomer provided similar data, where the ^1^*J*_CN_ coupling constants were higher than ^2^*J*_CN_ coupling constants, C-3′ (^1^*J*_C3′-N2′_ = 2.25 Hz), C-4′ (^2^*J*_C4′-N2′_ = 1.11 Hz), and C-5′ (^2^*J*_C5′-N2′_ = 1.52 Hz) in the 1,2-oxazole moiety, which is in good agreement with the data reported in the literature [73]. Moreover, the ^2^*J*_CN_ and ^3^*J*_CN_ couplings were observed for the signals from the pyrazole fragment. These ^13^C-^15^N spin–spin interactions with adjacent phenyl and pyrazole moieties were an additional criterion to confirm the final structures of the pyrazole-isoxazoles **16** and **17**.

## 3. Materials and Methods

### 3.1. General Information

All starting materials were purchased from commercial suppliers and were used as received. Microwave reactions were conducted using a CEM Discover synthesis unit (CEM Corp., Matthews, NC, USA) and performed in glass vessels (capacity: 10 mL), sealed with a septum. The pressure was controlled by a load cell connected to the vessel. The temperature of the contents of the vessel was monitored using a calibrated infrared temperature controller, mounted under the reaction vessel. All experiments were performed with stirring. Flash column chromatography was performed on silica gel, 60 Å (230–400 µm, Merck). Thin-layer chromatography was carried out on silica gel plates (Merck Kieselgel 60 F254) and visualized by UV light (254 nm). The melting points were determined on a Büchi M-565 melting point apparatus (Büchi Labortechnik AG, Flawil, Switzerland) and were uncorrected. The IR spectra were recorded on a Bruker Vertex 70v FT-IR spectrometer (Bruker Optik GmbH, Ettlingen, Germany) using neat samples or on a Bruker Tensor 27 (Bruker Optik GmbH, Ettlingen, Germany) spectrometer using KBr pellets and were reported in frequency of absorption (cm^−1^). Mass spectra were obtained on a Shimadzu LCMS-2020 (ESI^+^) spectrometer (Shimadzu Corporation, Kyoto, Japan). High-resolution mass spectra were measured on a Bruker MicrOTOF-Q III (ESI^+^) apparatus (Bruker Daltonik GmbH, Bremen, Germany). The ^1^H, ^13^C and ^15^N NMR spectra were recorded in CDCl_3_ or DMSO-*d*_6_ solutions at 25 °C on a Bruker Avance III 700 (700 MHz for ^1^H, 176 MHz for ^13^C, and 71 MHz for ^15^N) spectrometer (Bruker BioSpin AG, Fallanden, Switzerland), equipped with a 5 mm TCI ^1^H-^13^C/^15^N/D z-gradient cryoprobe and a Bruker Avance III 400 (400 MHz for ^1^H, 101 MHz for ^13^C, and 40 MHz for ^15^N) spectrometer (Bruker BioSpin AG), using a 5 mm directly detecting BBO probe. The chemical shifts (δ), expressed in ppm, were relative to tetramethylsilane (TMS). The ^15^N NMR spectra were referenced to neat, external nitromethane (coaxial capillary). Full and unambiguous assignment of the ^1^H, ^13^C and ^15^N NMR resonances was achieved using a combination of standard NMR spectroscopic techniques [74], such as DEPT, COSY, TOCSY, NOESY, ROESY, gs-HSQC, gs-HMBC, H2BC, LR-HSQMBC and 1,1-ADEQUATE experiments [75]. The following abbreviations are used in reporting NMR data: Ph, phenyl; Pz, pyrazole; Pyr, pyridine; Naph, naphtalene; Quin, quinoline; Th, thiophene; Ox, 1,2-oxazole. ^1^H-, ^13^C-, and ^1^H-^15^N HMBC NMR spectra, and HRMS data of the new compounds are provided in the Appendix A.

### 3.2. Chemistry

#### 3.2.1. General Procedure for the Synthesis of **2b,c**

To a solution of 1-phenyl-1*H*-pyrazol-3-ol (**1**) (320 mg, 2 mmol) in abs. DMF (20 mL), cooled to 0 °C under an inert atmosphere, NaH (60% dispersion in mineral oil, 80 mg, 2 mmol) was added portion wise [76]. After stirring the mixture for 15 min, iodopropane (for **2b**) or 1-chloro-2-methoxyethane (for **2c**) (2.4 mmol) was added dropwise. The reaction mixture was stirred at 60 °C for 1 h, poured into water (20 mL) and extracted with ethyl acetate (3 × 20 mL). The organic layers were combined, washed with brine, dried over Na_2_SO_4_, and filtrated, and the solvent was evaporated. The residue was purified by column chromatography (SiO_2_, eluent: ethyl acetate/*n*-hexane, 1:7, *v*/*v*) to produce pure **2b,c**.

##### 1-Phenyl-3-propoxy-1*H*-pyrazole (**2b**)

Colorless liquid; yield 64% (259 mg); *R_f_* = 0.43 (EtOAc/Hex 1/2, *v*/*v*). IR (*ν*_max_, cm^−1^): 3048, 2965, 1601, 1464, 1366, 1177, 1049, 935. ^1^H NMR (400 MHz, CDCl_3_): *δ*_H_ ppm 1.06 (t, *J* = 7.4 Hz, 3H, CH_3_), 1.80–1.89 (m, 2H, CH_3_CH_2_), 4.21 (t, *J* = 6.6 Hz, 2H, OCH_2_), 5.89 (d, *J* = 2.4 Hz, 1H, 4-H), 7.17–7.21 (m, 1H, Ph 4-H), 7.38–7.42 (m, 2H, Ph 3,5-H), 7.60–7.62 (m, 2H, Ph 2,6-H), 7.73 (d, *J* = 2.3 Hz, 1H, 5-H). ^13^C NMR (101 MHz, CDCl_3_): *δ*_C_ ppm 10.5 (CH_3_), 22.6 (CH_2_CH_3_), 70.8 (OCH_2_), 93.7 (C-4), 117.8 (Ph C-2,6), 125.2 (Ph C-4), 127.5 (C-5), 129.3 (Ph C-3,5), 140.3 (Ph C-1), 164.8 (C-3). HRMS (ESI^+^) for C_12_H_15_N_2_O ([M + H]^+^) calcd 203.1179, found 203.1181.

##### 3-(2-Methoxyethoxy)-1-phenyl-1*H*-pyrazole (**2c**)

Colorless liquid; yield 83% (362 mg); *R_f_* = 0.43 (EtOAc/Hex 1/3, *v*/*v*). IR (*ν*_max_, cm^−1^): 752, 1051, 1352, 1482, 1506, 1543, 1660, 2815, 2881, 2931, 2982, 3048, 3070, 3127, 3146. ^1^H NMR (700 MHz, CDCl_3_): *δ*_H_ ppm 3.45 (s, 3H, CH_3_), 3.75–3.75 (m, 2H, CH_3_OCH_2_CH_2_O), 4.42–4.45 (m, 2H, CH_3_OCH_2_CH_2_O), 5.93 (d, *J* = 2.6 Hz, 1H, 4-H), 7.18–7.20 (m, 1H, Ph 4-H), 7.37–7.43 (m, 2H, Ph 3,5-H), 7.58–7.60 (m, 2H, Ph 2,6-H), 7.72 (d, *J* = 2.6 Hz, 1H, 5-H). ^13^C NMR (176 MHz, CDCl_3_): *δ*_C_ ppm 59.2 (OCH_3_), 68.2 (CH_3_OCH_2_CH_2_O), 71.1 (CH_3_OCH_2_CH_2_O), 94.2 (C-4), 117.9 (Ph C-2,6), 125.3 (Ph C-4), 127.8 (C-5), 129.4 (Ph C-3,5), 140.3 (Ph C-1), 164.3 (C-3). HRMS (ESI^+^) for C_12_H_14_N_2_NaO_2_ ([M + Na]^+^) calcd 241.0947, found 241.0948.

#### 3.2.2. General Procedure for the Synthesis of **3b,c**

Phosphoryl chloride (0.37 mL, 4 mmol) was added dropwise to DMF (0.31 mL, 4 mmol) at −10 °C. Then, **2b,c** (1 mmol) was added to the Vilsmeier–Haack complex, and the reaction mixture was heated at 70 °C for 1 h. After the neutralization with 10% aq NaHCO_3_, the precipitate was filtered off and recrystallized from DCM to produce pure **3b,c**.

##### 1-Phenyl-3-propoxy-1*H*-pyrazole-4-carbaldehyde (**3b**)

Colorless solid; yield 84% (193 mg); m.p. 90–91 °C; *R_f_* = 0.38 (EtOAc/Hex 1/4, *v*/*v*). IR (*ν*_max_, cm^−1^): 3103, 2960, 1735, 1669, 1559, 1370, 1206, 1007, 867, 754. ^1^H NMR (400 MHz, CDCl_3_): *δ*_H_ ppm 1.06 (t, *J* = 7.4 Hz, 3H, CH_3_), 1.86–1.92 (m, 2H, CH_2_CH_3_), 4.36 (t, *J* = 6.7 Hz, 2H, OCH_2_), 7.30–7.34 (m, 1H, Ph 4-H), 7.44–7.48 (m, 2H, Ph 3,5-H), 7.63–7.65 (m, 2H, Ph 2,6-H), 8.25 (s, 1H, 5-H), 9.87 (s, 1H, CHO). ^13^C NMR (101 MHz, CDCl_3_): *δ*_C_ ppm 10.6 (CH_3_), 22.5 (CH_2_CH_3_), 71.2 (OCH_2_), 111.6 (C-4), 119.0 (Ph C-2,6), 127.4 (Ph C-4), 129.3 (C-5), 129.7 (Ph C-3,5), 139.2 (Ph C-1), 164.3 (C-3), 183.6 (CHO). HRMS (ESI^+^) for C_13_H_14_N_2_NaO_2_ ([M + Na]^+^) calcd 253.0947, found 253.0950.

##### 3-(2-Methoxyethoxy)-1-phenyl-1*H*-pyrazole-4-carbaldehyde (**3c**)

Colorless solid; yield 90% (221 mg); m.p. 91–92 °C; *R_f_* = 0.19 (EtOAc/Hex 1/3, *v*/*v*). IR (*ν*_max_, cm^−1^): 3121, 3099, 2975, 2812, 1752, 1669, 1500, 1225, 761.^1^H NMR (400 MHz, CDCl_3_): *δ*_H_ ppm 3.47 (s, 3H, CH_3_), 3.83 (t, *J* = 3.8 Hz, 2H, CH_3_OCH_2_), 4.57 (t, *J* = 3.8 Hz, 2H, CH_3_OCH_2_CH_2_), 7.31–7.35 (m, 1H, Ph 4-H), 7.45–7.48 (m, 2H, Ph 3,5-H), 7.63–7.65 (m, 2H, Ph 2,6-H), 8.26 (s, 1H, 5-H), 9.89 (s, 1H, CHO). ^13^C NMR (101 MHz, CDCl_3_): *δ*_C_ ppm 59.4 (CH_3_), 68.9 (CH_3_OCH_2_CH_2_), 70.8 (CH_3_OCH_2_), 111.6 (C-4), 119.0 (Ph C-2,6), 127.4 (Ph C-4), 129.3 (C-5), 129.8 (Ph C-3,5), 139.2 (Ph C-1), 164.0 (C-3), 183.7 (CHO). HRMS (ESI^+^) for C_13_H_14_N_2_NaO_3_ ([M + Na]^+^) calcd 269.0897, found 269.0897.

#### 3.2.3. General Procedure for the Synthesis of **4a**–**k**

To a solution of appropriate 3-(alkyloxy)-1-phenyl-1*H*-pyrazole-4-carbaldehyde (**3a**–**d**) (1 mmol) in EtOH (96%, 5 mL), NaOH (0.2 g, 5 mmol) and appropriate acetophenone (1.1 mmol) were added. The reaction mixture was stirred at 55 °C for 30 min., cooled to room temperature, diluted with H_2_O, and extracted with EtOAc (3 × 10 mL). The organic layers were combined, dried with sodium sulphate, filtrated off and concentrated. The obtained residue was purified by column chromatography (SiO_2_, eluent: ethyl acetate/*n*-hexane, 1:6, *v*/*v*) to provide the desired product **4a**–**k**.

##### (2*E*)-3-(3-Methoxy-1-phenyl-1*H*-pyrazol-4-yl)-1-phenylprop-2-en-1-one (**4a**)

Yellow solid; yield 61% (186 mg); m.p. 127–131 °C; *R_f_* = 0.47 (EtOAc/Hex 1/4, *v*/*v*). IR (*ν*_max_, KBr, cm^−1^): 3142, 3111, 3040, 2951, 1660 (C=O), 1599, 1592, 1506, 1416, 1219, 1022, 973, 746, 681, 638. ^1^H NMR (700 MHz, CDCl_3_): *δ*_H_ ppm 4.14 (s, 3H, OCH_3_), 7.24–7.27 (m, 1H, NPh 4-H), 7.42–7.46 (m, 2H, NPh 3,5-H), 7.48–7.51 (m, 2H, CPh 3,5-H), 7.55–7.57 (m, 1H, CPh 4-H), 7.60 (d, *J* = 15.5 Hz, 1H, CHCHC(O)Ph), 7.62–7.64 (m, 2H, NPh 2,6-H), 7.74 (d, *J* = 15.5 Hz, 1H, CHCHC(O)Ph), 7.96 (s, 1H, Pz 5-H), 8.02–8.03 (m, 2H, CPh 2,6-H). ^13^C NMR (176 MHz, CDCl_3_): *δ*_C_ ppm 56.6 (OCH_3_), 107.0 (Pz C-4), 118.1 (NPh C-2,6), 120.5 (CHCHC(O)Ph), 126.2 (NPh C-4), 128.4 (CPh C-2,6), 128.5 (CPh C-3,5), 129.0 (Pz C-5), 129.5 (NPh C-3,5), 132.4 (CPh C-4), 133.7 (CHCHC(O)Ph), 138.6 (CPh C-1), 139.4 (NPh C-1), 163.3 (Pz C-3), 190.7 (C=O). ^15^N NMR (71 MHz, CDCl_3_): *δ*_N_ ppm −183.7 (N-1), −118.7 (N-2). HRMS (ESI^+^) for C_19_H_16_N_2_NaO_2_ ([M + Na]^+^) calcd 327.1104, found 327.1101.

##### (2*E*)-1-(4-Methoxyphenyl)-3-(3-methoxy-1-phenyl-1*H*-pyrazol-4-yl)prop-2-en-1-one (**4b**)

Yellowish white solid; yield 73% (244 mg); m.p. 167–169 °C; *R_f_* = 0.22 (EtOAc/Hex 1/6, *v*/*v*). IR (*ν*_max_, KBr, cm^−1^): 3137, 3098, 3072, 2948, 1656 (C=O), 1510, 1422, 1253, 1227, 1176, 1020, 974, 833, 754, 689, 603. ^1^H NMR (700 MHz, CDCl_3_): *δ*_H_ ppm 3.88 (s, 3H, PhOCH_3_), 4.15 (s, 3H, PzOCH_3_), 6.97–6.99 (m, 2H, CPh 3,5-H), 7.25–7.27 (m, 1H, NPh 4-H), 7.43–7.45 (m, 2H, NPh 3,5-H), 7.61 (d, *J* = 15.5 Hz, 1H, CHCHC(O)Ph), 7.63–7.64 (m, 2H, NPh 2,6-H), 7.72 (d, *J* = 15.4 Hz, 1H, CHCHC(O)Ph), 7.96 (s, 1H, Pz 5-H), 8.04–8.05 (m, 2H, CPh 2,6-H). ^13^C NMR (176 MHz, CDCl_3_): *δ*_C_ ppm 55.6 (PhOCH_3_), 56.7 (PzOCH_3_), 107.3 (Pz C-4), 113.8 (CPh C-3,5), 118.2 (NPh C-2,6), 120.5 (CHCHC(O)Ph), 126.2 (NPh C-4), 129.0 (Pz C-5), 129.6 (NPh C-3,5), 130.8 (CPh C-2,6), 131.6 (CPh C-1), 133.0 (CHCHC(O)Ph), 139.6 (NPh C-1), 163.3 (CPh C-4), 163.4 (Pz C-3), 189.1 (C=O). ^15^N NMR (71 MHz, CDCl_3_): *δ*_N_ ppm −185.2 (N-1), −118.5 (N-2). HRMS (ESI^+^) for C_20_H_18_N_2_NaO_3_ ([M + Na]^+^) calcd 357.1210, found 357.1212.

##### (2*E*)-1-(4-Fluorophenyl)-3-(3-methoxy-1-phenyl-1*H*-pyrazol-4-yl)prop-2-en-1-one (**4c**)

Yellow solid; yield 60% (193 mg); m.p. 144–147 °C; *R_f_* = 0.3 (EtOAc/Hex 1/6, *v*/*v*). IR (*ν*_max_, KBr, cm^−1^): 3138, 3106, 3061, 2950, 1660 (C=O), 1585, 1514, 1504, 1421, 1218, 1014, 973, 826, 750, 591. ^1^H NMR (700 MHz, CDCl_3_): *δ*_H_ ppm 4.15 (s, 3H, CH_3_), 7.15–7.18 (m, 2H, CPh 3,5-H), 7.26–7.28 (m, 1H, NPh 4-H), 7.44–7.46 (m, 2H, NPh 3,5-H), 7.57 (d, *J* = 15.4 Hz, 1H, CHCHC(O)Ph), 7.63–7.65 (m, 2H, NPh 2,6-H), 7.74 (d, *J* = 15.6 Hz, 1H, CHCHC(O)Ph), 7.97 (s, 1H, Pz 5-H), 8.04–8.07 (m, 2H, CPh 2,6-H). ^13^C NMR (176 MHz, CDCl_3_): *δ*_C_ ppm 56.8 (CH_3_), 107.1 (Pz C-4), 115.7 (d, ^2^*J* = 21.8 Hz, CPh C-3,5), 118.2 (NPh C-2,6), 120.1 (CHCHC(O)Ph), 126.4 (NPh C-4), 129.2 (Pz C-5), 129.6 (NPh C-3,5), 131.1 (d, ^3^*J* = 9.2 Hz, CPh C-2,6), 134.1 (CHCHC(O)Ph), 135.0 (d, ^4^*J* = 3.1 Hz, CPh C-1), 139.5 (NPh C-1), 163.5 (Pz C-3), 165.6 (d, ^1^*J* = 253.6 Hz, CPh C-4), 189.1 (C=O). ^15^N NMR (71 MHz, CDCl_3_): *δ*_N_ ppm −184.3 (N-1), −118.5 (N-2). HRMS (ESI^+^) for C_19_H_15_FN_2_NaO_2_ ([M + Na]^+^) calcd 345.1010, found 345.1007.

##### (2*E*)-1-(4-Chlorophenyl)-3-(3-methoxy-1-phenyl-1*H*-pyrazol-4-yl)prop-2-en-1-one (**4d**)

Yellow solid; yield 66% (224 mg); m.p. 179–182 °C; *R_f_* = 0.37 (EtOAc/Hex 1/8, *v*/*v*). IR (*ν*_max_, KBr, cm^−1^): 3139, 3108, 3071, 2950, 1659 (C=O), 1592, 1514, 1505, 1421, 1219, 1013, 973, 823, 750, 685, 655. ^1^H NMR (700 MHz, CDCl_3_): *δ*_H_ ppm 4.15 (s, 3H, CH_3_), 7.26–7.29 (m, 1H, NPh 4-H), 7.44–7.48 (m, 4H, NPh 3,5-H, CPh 3,5-H), 7.55 (d, *J* = 15.4 Hz, 1H, CHCHC(O)Ph), 7.63–7.65 (m, 2H, NPh 2,6-H), 7.74 (d, *J* = 15.5 Hz, 1H, CHCHC(O)Ph), 7.96–7.97 (m, 3H, Pz 5-H, CPh 2,6-H). ^13^C NMR (176 MHz, CDCl_3_): *δ*_C_ ppm 56.8 (CH_3_), 107.1 (Pz C-4), 118.3 (NPh C-2,6), 120.1 (CHCHC(O)Ph), 126.4 (NPh C-4), 129.0 (CPh C-3,5), 129.3 (Pz C-5), 129.7 (NPh C-3,5), 130.0 (CPh C-2,6), 134.4 (CHCHC(O)Ph), 137.0 (CPh C-1), 138.9 (CPh C-4), 139.5 (NPh C-1), 163.5 (Pz C-3), 189.5 (C=O). ^15^N NMR (71 MHz, CDCl_3_): *δ*_N_ ppm −183.9 (N-1), −119.0 (N-2). HRMS (ESI^+^) for C_19_H_15_ClN_2_NaO_2_ ([M + Na]^+^) calcd 361.0714, found 361.0716.

##### (2*E*)-1-Phenyl-3-(1-phenyl-3-propoxy-1*H*-pyrazol-4-yl)prop-2-en-1-one (**4e**)

Yellow solid; yield 58% (156 mg); m.p. 133–135 °C; *R_f_* = 0.49 (EtOAc/Toluene 1/12, *v*/*v*). IR (*ν*_max_, KBr, cm^−1^): 3136, 3105, 3071, 2964, 1655 (C=O), 1588, 1574, 1502, 1417, 1215, 1021, 1010, 998, 748, 685, 643. ^1^H NMR (700 MHz, CDCl_3_): *δ*_H_ ppm 1.16 (t, *J* = 7.5 Hz, 3H, CH_3_), 1.93–1.98 (m, 2H, OCH_2_CH_2_CH_3_), 4.42 (t, *J* = 6.5 Hz, 2H, OCH_2_CH_2_CH_3_), 7.25–7.28 (m, 1H, NPh 4-H), 7.43–7.46 (m, 2H, NPh 3,5-H), 7.49–7.53 (m, 2H, CPh 3,5-H), 7.57–7.59 (m, 1H, CPh 4-H), 7.63–7.66 (m, 2H, NPh 2,6-H), 7.69 (d, *J* = 15.5 Hz, 1H, CHCHC(O)Ph), 7.78 (d, *J* = 15.5 Hz, 1H, CHCHC(O)Ph), 7.98 (s, 1H, Pz 5-H), 8.04–8.06 (m, 2H, CPh 2,6-H). ^13^C NMR (176 MHz, CDCl_3_): *δ*_C_ ppm 10.7 (CH_3_), 22.6 (OCH_2_CH_2_CH_3_), 71.0 (OCH_2_CH_2_CH_3_), 107.3 (Pz C-4), 118.1 (NPh C-2,6), 120.4 (CHCHC(O)Ph), 126.1 (NPh C-4), 128.4 (CPh C-2,6), 128.6 (CPh C-3,5), 128.8 (Pz C-5), 129.5 (NPh C-3,5), 132.5 (CPh C-4), 133.9 (CHCHC(O)Ph), 138.7 (CPh C-1), 139.5 (NPh C-1), 163.1 (Pz C-3), 190.6 (C=O). ^15^N NMR (71 MHz, CDCl_3_): *δ*_N_ ppm −184.4 (N-1), −119.3 (N-2). HRMS (ESI^+^) for C_21_H_20_N_2_NaO_2_ ([M + Na]^+^) calcd 355.1417, found 355.1416.

##### (2*E*)-3-[3-(2-Methoxyethoxy)-1-phenyl-1*H*-pyrazol-4-yl]-1-phenylprop-2-en-1-one (**4f**)

Yellow solid; yield 62% (216 mg); m.p. 116–117 °C; *R_f_* = 0.28 (EtOAc/Hex 1/6, *v*/*v*). IR (*ν*_max_, KBr, cm^−1^): 3138, 3101, 3063, 3040, 2931, 1655 (C=O), 1594, 1503, 1413, 1219, 1051, 976, 848, 743, 683, 640. ^1^H NMR (700 MHz, CDCl_3_): *δ*_H_ ppm 3.52 (s 3H, CH_3_), 3.87 (t, *J* = 4.6 Hz, 2H, OCH_2_CH_2_OCH_3_), 4.59 (t, *J* = 4.6 Hz, 2H, OCH_2_CH_2_OCH_3_), 7.25–7.27 (m, 1H, NPh 4-H), 7.43–7.45 (m, 2H, NPh 3,5-H), 7.48–7.50 (m, 2H, CPh 3,5-H), 7.55–7.57 (m, 1H, CPh 4-H), 7.61–7.63 (m, 2H, NPh 2,6-H), 7.71 (d, *J* = 14.6 Hz, 1H, CHCHC(O)Ph), 7.75 (d, *J* = 15.5 Hz, 1H, CHCHC(O)Ph), 7.97 (s, 1H, Pz 5-H), 8.03–8.05 (m, 2H, CPh 2,6-H). ^13^C NMR (176 MHz, CDCl_3_): *δ*_C_ ppm 59.3 (CH_3_), 68.7 (OCH_2_CH_2_OCH_3_), 71.1 (OCH_2_CH_2_OCH_3_), 107.4 (Pz C-4), 118.2 (NPh C-2,6), 120.8 (CHCHC(O)Ph), 126.3 (NPh C-4), 128.57 (CPh C-2,6), 128.63 (CPh C-3,5), 128.8 (Pz C-5), 129.6 (NPh C-3,5), 132.6 (CPh C-4), 133.6 (CHCHC(O)Ph), 138.7 (CPh C-1), 139.5 (NPh C-1), 162.8 (Pz C-3), 190.6 (C=O). ^15^N NMR (71 MHz, CDCl_3_): *δ*_N_ ppm −184.2 (N-1), −118.3 (N-2). HRMS (ESI^+^) for C_21_H_20_N_2_NaO_3_ ([M + Na]^+^) calcd 371.1366, found 371.1364.

##### (2*E*)-3-[3-(Benzyloxy)-1-phenyl-1*H*-pyrazol-4-yl]-1-phenylprop-2-en-1-one (**4g**)

Yellow solid; yield 66% (253 mg); m.p. 168–169 °C; *R_f_* = 0.36 (EtOAc/Hex 1/4, *v*/*v*). IR (*ν*_max_, cm^−1^): 3063, 3029, 1654 (C=O), 1591, 1567, 1504, 1359, 1005, 970, 779, 680. ^1^H NMR (700 MHz, CDCl_3_): *δ*_H_ ppm 5.50 (s, 2H, CH_2_), 7.27–7.29 (m, 1H, NPh 4-H), 7.39–7.41 (m, 1H, CH_2_Ph 4-H), 7.44–7.47 (m, 6H, CH_2_Ph 3,5-H, C(O)Ph 3,5-H, NPh 3,5-H), 7.54–7.56 (m, 1H, C(O)Ph 4-H), 7.58–7.59 (m, 2H, CH_2_Ph 2,6-H), 7.65–7.66 (m, 2H, NPh 2,6-H), 7.74 (d, *J* = 15.5 Hz, 1H, CHCHC(O)Ph), 7.77 (d, *J* = 15.4 Hz, 1H, CHCHC(O)Ph), 7.96–7.97 (m, 2H, C(O)Ph 2,6-H), 7.99 (s, 1H, Pz 5-H). ^13^C NMR (176 MHz, CDCl_3_): *δ*_C_ ppm 71.2 (CH_2_), 107.4 (Pz C-4), 118.3 (NPh C-2,6), 120.8 (CHCHC(O)Ph), 126.4 (NPh C-4), 128.0 (CH_2_Ph C-2,6), 128.3 (CH_2_Ph C-4), 128.5 (C(O)Ph C-2,6), 128.6 (C(O)Ph C-3,5), 128.7 (CH_2_Ph C-3,5), 129.0 (Pz C-5), 129.7 (NPh C-3,5), 132.6 (C(O)Ph C-4), 133.5 (CHCHC(O)Ph), 136.9 (CH_2_Ph C-1), 138.6 (C(O)Ph C-1), 139.5 (NPh C-1), 162.8 (Pz C-3), 190.4 (C=O). ^15^N NMR (71 MHz, CDCl_3_): *δ*_N_ ppm −184.29 (Pz N-1), −117.99 (Pz N-2). HRMS (ESI^+^) for C_25_H_20_N_2_NaO_2_ ([M + Na]^+^) calcd 403.1417, found 403.1416.

##### (2*E*)-3-[3-(Benzyloxy)-1-phenyl-1*H*-pyrazol-4-yl]-1-(4-methoxyphenyl)prop-2-en-1-one (**4h**)

Yellow solid; yield 68% (279 mg); m.p. 164–165 °C; *R_f_* = 0.34 (EtOAc/Hex 1/4, *v*/*v*). IR (*ν*_max_, cm^−1^): 3064, 2928, 1652 (C=O), 1502, 1417, 1221, 1169, 1010, 972, 826, 700. ^1^H NMR (700 MHz, CDCl_3_): *δ*_H_ ppm 3.88 (s, 3H, CH_3_), 5.50 (s, 2H, CH_2_), 6.92–6.93 (m, 2H, C(O)Ph 2,6-H), 7.27–7.28 (m, 1H, NPh 4-H), 7.39–7.41 (m, 1H, CH_2_Ph 4-H), 7.44–7.47 (m, 4H, CH_2_Ph 3,5-H, NPh 3,5-H), 7.58–7.60 (m, 2H, CH_2_Ph 2,6-H), 7.64–7.66 (m, 2H, NPh 2,6-H), 7.74 (s, 2H, CHCHC(O)Ph), 7.96–7.97 (m, 2H, C(O)Ph 3,5-H), 7.98 (s, 1H, Pz 5-H). ^13^C NMR (176 MHz, CDCl_3_): *δ*_C_ ppm 55.6 (CH_3_), 71.2 (CH_2_), 107.5 (Pz C-4), 113.8 (C(O)Ph C-2,6), 118.2 (NPh C-2,6), 120.7 (CHCHC(O)Ph), 126.3 (NPh C-4), 128.0 (CH_2_Ph C-2,6), 128.3 (CH_2_Ph C-4), 128.7 (CH_2_Ph C-3,5), 128.9 (Pz C-5), 129.7 (NPh C-3,5), 130.8 (C(O)Ph C-3,5), 131.6 (C(O)Ph C-1), 132.7 (CHCHC(O)Ph), 137.0 (CH_2_Ph C-1), 139.6 (NPh C-1), 162.7 (Pz C-3), 163.3 (C(O)Ph C-4), 188.8 (C=O). ^15^N NMR (71 MHz, CDCl_3_): *δ*_N_ ppm −184.44 (Pz N-1), −117.93 (Pz N-2). HRMS (ESI^+^) for C_26_H_22_N_2_NaO_3_ ([M + Na]^+^) calcd 433.1523, found 433.1525.

##### (2*E*)-3-[3-(Benzyloxy)-1-phenyl-1*H*-pyrazol-4-yl]-1-(4-chlorophenyl)prop-2-en-1-one (**4i**)

Yellow solid; yield 97% (411 mg); m.p. 192–193 °C; *R_f_* = 0.56 (EtOAc/Hex 1/4, *v*/*v*). IR (*ν*_max_, cm^−1^): 3033, 2916, 1600 (C=O), 1501, 1409, 1365, 1215, 1003, 973, 823, 740. ^1^H NMR (700 MHz, CDCl_3_): *δ*_H_ ppm 5.49 (s, 2H, CH_2_), 7.28–7.30 (m, 1H, NPh 4-H), 7.40–7.42 (m, 3H, CH_2_Ph 4-H, 4ClPh 2,6-H), 7.44–7.48 (m, 4H, CH_2_Ph 3,5-H, NPh 3,5-H), 7.57–7.58 (m, 2H, CH_2_Ph 2,6-H), 7.65–7.66 (m, 2H, NPh 2,6-H), 7.68 (d, *J* = 15.4 Hz, 1H, CHCHC(O)Ph), 7.77 (d, *J* = 15.4 Hz, 1H, CHCHC(O)Ph), 7.87–7.89 (m, 2H, 4ClPh 3,5-H), 7.99 (s, 1H, Pz 5-H). ^13^C NMR (176 MHz, CDCl_3_): *δ*_C_ ppm 71.3 (CH_2_), 107.3 (Pz C-4), 118.3 (NPh C-2,6), 120.1 (CHCHC(O)Ph), 126.5 (NPh C-4), 128.1 (CH_2_Ph C-2,6), 128.4 (CH_2_Ph C-4), 128.7 (4ClPh C-2,6), 128.9 (CH_2_Ph C-3,5), 129.1 (Pz C-5), 129.7 (NPh C-3,5), 129.9 (4ClPh C-3,5), 134.1 (CHCHC(O)Ph), 136.9 (CH_2_Ph C-1), 137.0 (4ClPh C-4), 139.0 (4ClPh C-1), 139.5 (NPh C-1), 162.8 (Pz C-3), 189.0 (C=O). ^15^N NMR (71 MHz, CDCl_3_): *δ*_N_ ppm −183.94 (Pz N-1), −117.85 (Pz N-2). HRMS (ESI^+^) for C_25_H_19_ClN_2_NaO_3_ ([M + Na]^+^) calcd 437.1027, found 437.1028.

##### (2*E*)-3-[3-(Benzyloxy)-1-phenyl-1*H*-pyrazol-4-yl]-1-(4-fluorophenyl)prop-2-en-1-one (**4j**)

Yellow solid; yield 79% (318 mg); m.p. 145–146 °C; *R_f_* = 0.54 (EtOAc/Hex 1/4, *v*/*v*). IR (*ν*_max_, cm^−1^): 3033, 2944, 1654 (C=O), 1596, 1500, 1411, 1364, 1002, 973, 826, 685. ^1^H NMR (700 MHz, CDCl_3_): *δ*_H_ ppm 5.49 (s, 2H, CH_2_), 7.09–7.12 (m, 2H, 4FPh 3,5-H), 7.27–7.30 (m, 1H, NPh 4-H), 7.41–7.42 (m, 1H, CH_2_Ph 4-H), 7.44–7.48 (m, 4H, CH_2_Ph 3,5-H, NPh 3,5-H), 7.58–7.59 (m, 2H, CH_2_Ph 2,6-H), 7.65–7.66 (m, 2H, NPh 2,6-H), 7.70 (d, *J* = 15.4 Hz, 1H, CHCHC(O)Ph), 7.76 (d, *J* = 15.4 Hz, 1H, CHCHC(O)Ph), 7.95–7.98 (m, 2H, 4FPh 2,6-H), 7.99 (s, 1H, Pz 5-H). ^13^C NMR (176 MHz, CDCl_3_): *δ*_C_ ppm 71.3 (CH_2_), 107.3 (Pz C-4), 115.7 (d, ^2^*J* = 21.7 Hz, 4FPh C-3,5), 118.3 (NPh C-2,6), 120.3 (CHCHC(O)Ph), 126.4 (NPh C-4), 128.1 (CH_2_Ph C-2,6), 128.4 (CH_2_Ph C-4), 128.7 (CH_2_Ph C-3,5), 129.1 (Pz C-5), 129.7 (NPh C-3,5), 131.1 (d, ^3^*J* = 9.1 Hz, 4FPh C-2,6), 133.7 (CHCHC(O)Ph), 135.0 (d, ^4^*J* = 3.0 Hz, 4FPh C-1), 136.9 (CH_2_Ph C-1), 139.5 (NPh C-1), 162.8 (Pz C-3), 164.2 (d, ^1^*J* = 253.7 Hz, 4FPh C-4), 188.7 (C=O). ^15^N NMR (71 MHz, CDCl_3_): *δ*_N_ ppm −184.26 (Pz N-1), −118.00 (Pz N-2). HRMS (ESI^+^) for C_25_H_19_FN_2_NaO_3_ ([M + Na]^+^) calcd 421.1323, found 421.1323.

##### (2*E*)-3-[3-(Benzyloxy)-1-phenyl-1*H*-pyrazol-4-yl]-1-[4-(dimethylamino)phenyl]prop-2-en-1-one (**4k**)

Yellow solid; yield 85% (376 mg); m.p. 179–180 °C; *R_f_* = 0.15 (EtOAc/Hex 1/4, *v*/*v*). IR (*ν*_max_, cm^−1^): 3102, 2918, 1564 (C=O), 1434, 1404, 1358, 1166, 1028, 977, 809, 742. ^1^H NMR (700 MHz, CDCl_3_): *δ*_H_ ppm 3.06 (s, 6H, CH_3_), 5.50 (s, 2H, CH_2_), 6.64–6.66 (m, 2H, (CH_3_)_2_NPh 3,5-H), 7.24–7.26 (m, 1H, NNPh 4-H), 7.38–7.40 (m, 1H, CH_2_Ph 4-H), 7.43–7.46 (m, 4H, CH_2_Ph 3,5-H, NNPh 3,5-H), 7.60–7.61 (m, 2H, CH_2_Ph 2,6-H), 7.64–7.65 (m, 2H, NNPh 2,6-H), 7.72 (d, *J* = 15.4 Hz, 1H, CHCHC(O)Ph), 7.78 (d, *J* = 15.4 Hz, 1H, CHCHC(O)Ph), 7.94–7.96 (m, 3H, (CH_3_)_2_NPh 2,6-H, Pz 5-H). ^13^C NMR (176 MHz, CDCl_3_): *δ*_C_ ppm 40.2 (CH_3_), 71.0 (CH_2_), 107.7 (Pz C-4), 110.9 ((CH_3_)_2_NPh C-3,5), 118.1 (NNPh C-2,6), 121.1 (CHCHC(O)Ph), 126.1 (NNPh C-4), 126.4 ((CH_3_)_2_NPh C-1), 127.9 (CH_2_Ph C-2,6), 128.1 (CH_2_Ph C-4), 128.5 (Pz C-5), 128.6 (CH_2_Ph C-3,5), 129.6 (NNPh C-4), 130.8 ((CH_3_)_2_NPh C-2,6), 131.2 (CHCHC(O)Ph), 137.1 (CH_2_Ph C-1), 139.6 (NNPh C-1), 153.3 ((CH_3_)_2_NPh C-4), 162.6 (Pz C-3), 188.0 (C=O). ^15^N NMR (71 MHz, CDCl_3_): *δ*_N_ ppm −325.44 (N(CH_3_)_2_), −185.23 (Pz N-1), −118.32 (Pz N-2). HRMS (ESI^+^) for C_27_H_25_N_3_NaO_2_ ([M + Na]^+^) calcd 446.1839, found 446.1842.

#### 3.2.4. General Procedure for the Synthesis of **5a**–**c**

To a solution of **4****g**–**j** (1 mmol) in toluene (3 mL), TFA (3mL) was added. The reaction mixture was stirred at room temperature for 24 h. Toluene and trifluoroacetic acid were evaporated. The residue was recrystalized from ACN to produce pure **5a**–**c**.

##### (2*E*)-3-(3-Hydroxy-1-phenyl-1*H*-pyrazol-4-yl)-1-phenylprop-2-en-1-one (**5a**)

Yellow solid; yield 60% (208 mg); m.p. 257–258 °C; *R_f_* = 0.2 (EtOAc/Hex 1/4, *v*/*v*). IR (*ν*_max_, cm^−1^): 2922, 2852, 1655 (C=O), 1593, 1573, 1504, 1416, 1020, 974, 751, 684. ^1^H NMR (700 MHz, DMSO-*d*_6_): *δ*_H_ ppm 7.27–7.29 (m, 1H, NPh 4-H), 7.49–7.51 (m, 2H, NPh 3,5-H), 7.56–7.59 (m, 2H, CPh 3,5-H), 7.64–7.66 (m, 3H, CPh 4-H, CHCHC(O)Ph), 7.73–7.74 (m, 2H, NPh 2,6-H), 7.99–8.00 (m, 2H, CPh 2,6-H), 8.89 (s, 1H, Pz 5-H), 11.48 (s, 1H, OH). ^13^C NMR (176 MHz, DMSO DMSO-*d*_6_): *δ*_C_ ppm 106.5 (Pz C-4), 117.3 (NPh C-2,6), 118.8 (CHCHC(O)Ph), 125.9 (NPh C-4), 128.0 (CPh C-2,6), 128.8 (CPh C-3,5), 129.5 (Pz C-5), 129.6 (NPh C-3,5), 132.7 (CPh C-4), 134.3 (CHCHC(O)Ph), 138.1 (CPh C-1), 139.0 (NPh C-1), 161.9 (Pz C-3), 189.0 (C=O). ^15^N NMR (71 MHz, DMSO-*d*_6_): *δ*_N_ ppm −183.51 (Pz N-1). HRMS (ESI^+^) for C_18_H_14_N_2_NaO_2_ ([M + Na]^+^) calcd 313.0947, found 313.0948.

##### (2*E*)-3-(3-Hydroxy-1-phenyl-1*H*-pyrazol-4-yl)-1-(4-methoxyphenyl)prop-2-en-1-one (**5b**)

Yellow solid; yield 63% (202 mg); m.p. 250–251 °C; *R_f_* = 0.13 (EtOAc/Hex 1/4, *v*/*v*). IR (*ν*_max_, cm^−1^): 3097, 2935, 1655 (C=O), 1593, 1504, 1246, 1173, 1023, 972, 829, 753. ^1^H NMR (700 MHz, DMSO-*d*_6_): *δ*_H_ ppm 3.86 (s, 3H, CH_3_), 7.09–7.10 (m, 2H, CPh 2,6-H), 7.27–7.29 (m, 1H, NPh 4-H), 7.49–7.51 (m, 2H, NPh 3,5-H), 7.62 (d, *J* = 15.4 Hz, 1H, CHCHC(O)Ph), 7.67 (d, *J* = 15.4 Hz, 1H, CHCHC(O)Ph), 7.72–7.73 (m, 2H, NPh 2,6-H), 8.00–8.01 (m, 2H, CPh 3,5-H), 8.87 (s, 1H, Pz 5-H), 11.43 (s, 1H, OH). ^13^C NMR (176 MHz, DMSO-*d*_6_): *δ*_C_ ppm 55.5 (CH_3_), 106.6 (Pz C-4), 114.1 (CPh C-2,6), 117.3 (NPh C-2,6), 118.7 (CHCHC(O)Ph), 125.8 (NPh C-4), 129.3 (Pz C-5), 129.6 (CPh C-3,5), 130.3 (CPh C-3,5), 130.8 (CPh C-1), 133.2 (CHCHC(O)Ph), 139.1 (NPh C-1), 161.9 (Pz C-3), 162.9 (CPh C-4), 187.1 (C=O). ^15^N NMR (71 MHz, DMSO-*d*_6_): *δ*_N_ ppm −183.85 (Pz N-1), −118.86 (Pz N-2). HRMS (ESI^+^) for C_19_H_16_N_2_NaO_3_ ([M + Na]^+^) calcd 343.1053, found 343.1054.

##### (2*E*)-1-(4-Chlorophenyl)-3-(3-hydroxy-1-phenyl-1*H*-pyrazol-4-yl)prop-2-en-1-one (**5c**)

Yellow solid; yield 83% (270 mg); m.p. 290–291 °C; *R_f_* = 0.26 (EtOAc/Hex 1/4, *v*/*v*). IR (*ν*_max_, cm^−1^): 3068, 2928, 1655 (C=O), 1509, 1416, 1217, 1093, 1040, 971, 827, 742. ^1^H NMR (700 MHz, DMSO-*d*_6_): *δ*_H_ ppm 7.28–7.30 (m, 1H, NPh 4-H), 7.49–7.51 (m, 2H, NPh 3,5-H), 7.61–7.68 (m, 4H, CPh 3,5-H, CHCHC(O)Ph), 7.72–7.73 (m, 2H, NPh 2,6-H), 8.00–8.01 (m, 2H, CPh 2,6-H), 8.89 (s, 1H, Pz 5-H), 11.51 (s, 1H, OH). ^13^C NMR (176 MHz, DMSO-*d*_6_): *δ*_C_ ppm 107.0 (Pz C-4), 117.8 (NPh C-2,6), 118.8 (CHCHC(O)Ph), 126.4 (NPh C-4), 129.4 (CPh C-3,5), 130.1 (NPh C-3,5), 130.1 (Pz C-5), 130.3 (CPh C-2,6), 135.2 (CHCHC(O)Ph), 137.2 (CPh C-4), 138.1 (CPh C-1), 139.4 (NPh C-1), 162.4 (Pz C-3), 188.2 (C=O). ^15^N NMR (71 MHz, DMSO-*d*_6_): *δ*_N_ ppm −183.15 (Pz N-1), −118.69 (Pz N-2). HRMS (ESI^+^) for C_18_H_13_ClN_2_NaO_2_ ([M + Na]^+^) calcd 347.0558, found 347.0557.

#### 3.2.5. General Procedure for the Synthesis of **8a**–**l**

To a solution of 1-(3-hydroxy-1-phenyl-1*H*-pyrazol-4-yl)ethan-1-one (**7**) (2.02 g, 10 mmol) in EtOH (96%, 80 mL), NaOH (2 g, 50 mmol) and appropriate aldehyde (20 mmol) were added. The reaction mixture was stirred at 55 °C for 3–5 h, cooled to room temperature and neutralized to pH = 7 using 6N HCl. The precipitate was filtered off, washed with water and cold ether and recrystallized from ACN to produce pure **8a**–**l**.

##### (2*E*)-1-(3-Hydroxy-1-phenyl-1*H*-pyrazol-4-yl)-3-phenylprop-2-en-1-one (**8a**)

The reaction mixture was stirred for 3 h. White solid; yield 88% (2.56 g); m.p. 222–223 °C; *R_f_*=0.24 (EtOAc/Hex 1/4, *v*/*v*). IR (*ν*_max_, cm^−1^): 3075, 3058, 3026, 1654 (C=O), 1584, 1511, 1448, 1217, 1062, 746, 735, 693, 678. ^1^H NMR (700 MHz, DMSO-*d*_6_): *δ*_H_ ppm 7.32–7.34 (m, 1H, NPh 4-H), 7.44–7.45 (m, 1H, CPh 4-H), 7.46–7.48 (m, 2H, CPh 3,5-H), 7.51–7.53 (m, 2H, NPh 3,5-H), 7.69 (d, *J* = 15.7 Hz, 1H, C(O)CHCHPh), 7.75 (d, *J* = 15.6 Hz, 1H, C(O)CHCHPh), 7.76–7.78 (m, 2H, CPh 2,6-H), 7.84–7.85 (m, 2H, NPh 2,6-H), 9.16 (s, 1H, Pz- 5-H), 11.22 (s, 1H, OH). ^13^C NMR (176 MHz, DMSO-*d*_6_): *δ*_C_ ppm 111.1 (Pz- C-4), 118.0 (NPh C-2,6), 124.3 (C(O)CHCH), 126.5 (NPh C-4), 128.4 (CPh C-2,6), 129.0 (CPh C-3,5), 129.6 (NPh C-3,5), 130.4 (CPh C-4), 131.7 (Pz C-5), 134.7 (CPh C-1), 138.8 (NPh C-1), 141.4 (C(O)CHCH), 162.0 (Pz C-3), 182.6 (C=O). ^15^N NMR (71 MHz, DMSO-*d*_6_): δ_N_ ppm −181.9 (Pz N-1), −117.0 (Pz N-2). HRMS (ESI^+^) for C_18_H_14_N_2_O_2_ ([M + Na]^+^) calcd 313.0948, found 313.0947.

##### (2*E*)-3-(4-Fluorophenyl)-1-(3-hydroxy-1-phenyl-1*H*-pyrazol-4-yl)prop-2-en-1-one (**8b**)

The reaction mixture was stirred for 4.5 h. Orange solid; yield 70% (2.16 g); m.p. 239–240 °C; *R_f_* = 0.46 (DCM/MeOH 100/1, *v*/*v*). IR (*ν*_max_, cm^−1^): 3360, 3111, 2978, 1646 (C=O), 1583, 1505, 1359, 1326, 821, 747, 673, 505. ^1^H NMR (700 MHz, DMSO-*d*_6_): *δ*_H_ ppm 7.31–7.35 (m, 3H, NPh 4-H, CPh 3,5-H), 7.51–7.53 (m, 2H, NPh 3,5-H), 7.68 (s, 2H, C(O)CHCH), 7.83–7.86 (m, 4H, NPh 2,6-H, CPh 2,6-H), 9.14 (s, 1H, Pz 5-H), 11.20 (s, 1H, OH). ^13^C NMR (176 MHz, DMSO-*d*_6_): *δ*_C_ ppm 111.1 (Pz C-4), 116.0 (d, ^2^*J* = 21.8 Hz, CPh C-3,5), 118.0 (NPh C-2,6), 124.2 (C(O)CHCH), 126.6 (NPh C-4), 129.6 (NPh C-3,5), 130.7 (d, ^3^*J* = 8.6 Hz, CPh C-2,6), 131.4 (d, ^4^*J* = 2.9 Hz, CPh C-1), 131.7 (Pz C-5), 138.8 (NPh C-1), 140.1 (C(O)CHCH), 161.9 (Pz C-3), 163.2 (d, ^1^*J* = 248.6 Hz, CPh C-4), 182.5 (C=O). ^15^N NMR (71 MHz, DMSO-*d*_6_): δ_N_ ppm −182.1 (Pz N-1), −119.5 (Pz N-2). HRMS (ESI^+^) for C_18_H_14_N_2_O_2_ ([M + H]^+^) calcd 331.0853, found 331.0853.

##### (2*E*)-3-(4-Chlorophenyl)-1-(3-hydroxy-1-phenyl-1*H*-pyrazol-4-yl)prop-2-en-1-one (**8c**)

The reaction mixture was stirred for 4 h. Yellow solid; yield 77% (2505 mg); m.p. 354–355 °C; *R_f_* = 0.17 (EtOAc/Hex 1/4, *v*/*v*). IR (*ν*_max_, cm^−1^): 3110, 3071, 1654 (C=O), 1586, 1511, 1456, 1325, 1218, 1094, 1062, 815, 745, 709, 681, 497. ^1^H NMR (700 MHz, DMSO-*d*_6_) *δ*_H_ ppm 7.33–7.35 (m, 1H, NPh 4-H), 7.51–7.55 (m, 4H, NPh 3,5-H, CPh 3,5-H), 7.66 (d, *J* = 15.7 Hz, 1H, C(O)CHCHPh), 7.73 (d, *J* = 15.7 Hz, 1H, C(O)CHCHPh), 7.80–7.81 (m, 2H, CPh 2,6-H), 7.83–7.84 (m, 2H, NPh 2,6-H), 9.15 (s, 1H, Pz- 5-H), 11.22 (s, 1H, OH). ^13^C NMR (176 MHz, DMSO-*d*_6_) *δ*_C_ ppm 111.5 (Pz C-4), 118.5 (NPh C-2,6), 125.4 (C(O)CHCHPh), 127.1 (NPh C-4), 129.5 (CPh C-3,5), 130.0 (NPh C-3,5), 130.6 (CPh C-2,6), 132.3 (Pz 5-H), 134.2, 135.3, 139.3 (NPh C-1), 140.4 (C(O)CHCHPh), 162.3 (Pz C-3), 182.8 (C=O). ^15^N NMR (71 MHz, DMSO-*d*_6_): δ_N_ ppm −181.7 (Pz N-1), −118.1 (Pz N-2). HRMS (ESI^+^) for C_18_H_13_ClN_2_O_2_ ([M + Na]^+^) calcd 347.0558, found 347.0558.

##### (2*E*)-1-(3-Hydroxy-1-phenyl-1*H*-pyrazol-4-yl)-3-(4-methoxyphenyl)prop-2-en-1-one (**8d**)

The reaction mixture was stirred for 3.5 h. Brown solid; yield 80% (2.57 g); m.p. 200–201 °C; *R_f_* = 0.14 (EtOAc/Hex 1/4, *v*/*v*). IR (*ν*_max_, cm^−1^): 3110, 3071, 1653 (C=O), 1586, 1509, 1457, 1219, 1172, 1049, 818, 769, 743, 679. ^1^H NMR (700 MHz, DMSO-*d*_6_): *δ*_H_ ppm 3.82 (s, 3H, CH_3_), 7.03–7.04 (m, 2H, CPh 3,5-H), 7.32–7.34 (m, 1H, NPh 4-H), 7.51–7.53 (m, 2H, NPh 3,5-H), 7.60 (d, *J* = 15.6 Hz, 1H, C(O)CHCH), 7.66 (d, *J* = 15.6 Hz, 1H, C(O)CHCH), 7.73–7.74 (m, 2H, CPh 2,6-H), 7.84–7.85 (m, 2H, NPh 2,6-H), 9.14 (s, 1H, Pz 5-H), 11.11 (s, 1H, OH). ^13^C NMR (176 MHz, DMSO-*d*_6_): *δ*_C_ ppm 55.4 (CH_3_), 111.0 (Pz C-4), 114.5 (CPh C-3,5), 118.0 (NPh C-2,6), 121.7 (C(O)CHCH), 126.5 (NPh C-4), 127.3 (CPh C-1), 129.6 (NPh C-3,5), 130.3 (CPh C-2,6), 131.5 (Pz C-5), 138.8 (NPh C-1), 141.4 (C(O)CHCH), 161.2 (Pz C-3), 161.9 (CPh C-4), 182.8 (C=O). ^15^N NMR (71 MHz, DMSO-*d*_6_): *δ*_N_ ppm −182.2 (Pz N-1). HRMS (ESI^+^) for C_19_H_16_N_2_O ([M + Na]^+^) calcd 343.1053, found 343.1053.

##### (2*E*)-1-(3-Hydroxy-1-phenyl-1*H*-pyrazol-4-yl)-3-[4-(trifluoromethoxy)phenyl]prop-2-en-1-one (**8e**)

The reaction mixture was stirred for 5 h. Yellow solid; yield 58% (2.21 g); m.p. 148–149 °C; *R_f_* = 0.49 (DCM/MeOH 100/1, *v*/*v*). IR (*ν*_max_, cm^−1^): 3110, 3071, 1657 (C=O), 1599, 1583, 1525, 1506, 1214, 1146, 977, 925, 825, 745. ^1^H NMR (400 MHz, DMSO-*d*_6_): *δ*_H_ ppm 7.32–7.36 (m, 1H, NPh 4-H), 7.45–7.47 (m, 2H, CPh 2,6-H), 7.50–7.54 (m, 2H, NPh 3,5-H), 7.69 (d, *J* = 15.8 Hz, 1H, C(O)CHCH), 7.74 (d, *J* = 15.8 Hz, 1H, C(O)CHCH), 7.83–7.85 (m, 2H, CPh 3,5-H), 7.90–7.92 (m, 2H, NPh 2,6-H), 9.15 (s, 1H, Pz 5-H), 11.24 (s, 1H, OH). ^13^C NMR (101 MHz, DMSO-*d*_6_): *δ*_C_ ppm 111.0 (Pz C-4), 118.1 (NPh C-2,6), 120.0 (q, ^1^*J* = 256.8 Hz, OCF_3_), 121.4, 125.4 (C(O)CHCHPh), 126.6, 129.6, 130.3, 131.8, 134.1 (Pz C-5), 138.8 (NPh C-1), 139.6 (C(O)CH), 149.4 (CPh C-4), 161.9 (Pz C-3), 182.4 (C=O). HRMS (ESI^+^) for C_19_H_13_F_3_N_2_O_3_ ([M + Na]^+^) calcd 397.0770, found 397.0770.

##### (2*E*)-3-[4-(Dimethylamino)phenyl]-1-(3-hydroxy-1-phenyl-1*H*-pyrazol-4-yl)prop-2-en-1-one (**8f**)

The reaction mixture was stirred for 5 h. Dark red solid; yield 25% (835 mg); m.p. 203–204 °C; *R_f_* = 0.09 (EtOAc/Hex 1/4, *v*/*v*). IR (*ν*_max_, cm^−1^): 3111, 1635 (C=O), 1586, 1505, 1426, 1354, 1160, 1034, 808, 750, 687, 668. ^1^H NMR (700 MHz, DMSO-*d*_6_): *δ*_H_ ppm 2.99 (s, 6H, CH_3_), 6.75–6.76 (m, 2H, CPh 3,5-H), 7.31–7.33 (m, 1H, NPh 4-H), 7.47 (d, d, *J* = 15.5 Hz, 1H, C(O)CH), 7.50–7.52 (m, 2H, NPh 3,5-H), 7.59–7.61 (m, 2H, CPh 3,5-H), 7.63 (d, d, *J* = 15.5 Hz, 1H, C(O)CHCH), 7.83–7.84 (m, 2H, NPh 2,6-H), 9.10 (s, 1H, Pz 5-H), 11.02 (s, 1H, OH). ^13^C NMR (176 MHz, DMSO-*d*_6_): *δ*_C_ ppm 39.7 (CH_3_), 111.0 (Pz C-4), 111.8 (CPh 3,5-C), 118.0 (NPh C-2,6), 118.3 (C(O)CHCH), 121.9 (CPh C-1), 126.4 (NPh C-4), 129.6 (CPh C-2,6), 130.3 (NPh C-3,5), 131.0 (Pz C-5), 138.9 (NPh C-1), 142.7 (C(O)CHCH), 151.9 (CPh C-4), 162.1 (Pz C-3), 183.1 (C=O). HRMS (ESI^+^) for C_20_H_19_N_3_O_2_ ([M + Na]^+^) calcd 356.1369, found 356.1369.

##### (2*E*)-1-(3-Hydroxy-1-phenyl-1*H*-pyrazol-4-yl)-3-(naphthalen-2-yl)prop-2-en-1-one (**8g**)

The reaction mixture was stirred for 3.5 h. Orange solid; yield 94% (3.20 g); m.p. 257–258 °C; *R_f_* = 0.43 (DCM/MeOH 100/1, *v*/*v*). IR (*ν*_max_, cm^−1^): 3117, 3056, 1650 (C=O), 1586, 1509, 460, 1322, 1216, 1062, 847, 806, 754, 732, 680. ^1^H NMR (700 MHz, DMSO-*d*_6_): *δ*_H_ ppm 7.34–7.36 (m, 1H, NPh 4-H), 7.52–7.55 (m, 2H, NPh 3,5-H), 7.57–7.60 (m, 2H, Naph 4,8-H), 7.85–7.87 (m, 4H, NPh 2,6-H, C(O)CHCH), C(O)CHCH), 7.96–8.02 (m, 4H, Naph 3,5,6,7-H), 8.26 (s, 1H, Naph 1-H), 9.21 (s, 1H, Pz 5-H), 11.19 (s, 1H, OH). ^13^C NMR (176 MHz, DMSO-*d*_6_): *δ*_C_ ppm 111.1 (Pz C-4), 118.1 (NPh C-2,6), 123.8 (Naph C-6), 124.5 (C(O)CHCH), 126.6 (NPh C-4), 126.9 (Naph C-4), 127.4 (Naph C-8), 127.8 (Naph C-3), 128.5 (Naph C-5), 128.6 (Naph C-7), 129.6 (NPh C-3,5), 130.4 (Naph C-1), 131.8 (Pz C-5), 132.3 (Naph C-4a), 133.0 (Naph C-2), 133.8 (Naph C-8a), 138.8 (NPh C-1), 141.4 (C(O)CHCH), 162.0 (Pz C-3), 182.6 (C=O). ^15^N NMR (71 MHz, DMSO-*d*_6_): *δ*_N_ ppm −181.9 (Pz N-1). HRMS (ESI^+^) for C_22_H_16_N_2_O_2_ ([M + Na]^+^) calcd 363.1104, found 363.1104.

##### (2*E*)-1-(3-Hydroxy-1-phenyl-1*H*-pyrazol-4-yl)-3-(pyridin-4-yl)prop-2-en-1-one (**8h**)

The reaction mixture was stirred for 4 h. Yellow solid; yield 36% (1.05 g); m.p. 230–231 °C; *R_f_* = 0.19 (DCM/MeOH 100/3, *v*/*v*). IR (*ν*_max_, cm^−1^): 3397, 3115, 3068, 1658 (C=O), 1585, 1509, 1453, 1318, 1216, 808, 748, 720, 675. ^1^H NMR (700 MHz, DMSO-*d*_6_): *δ*_H_ ppm 7.33–7.36 (m, 1H, Ph 4-H), 7.52–7.54 (m, 2H, Ph 3,5-H), 7.62 (d, *J* = 15.8 Hz, 1H, C(O)CHCH), 7.73–7.74 (m, 2H, Pyr 3,5-H), 7.84–7.85 (m, 2H, Ph 2,6-H), 7.90 (d, *J* = 15.8 Hz, 1H, C(O)CHCH), 8.68–8.69 (m, 2H, Ph 2,6-H), 9.19 (s, 1H, Pz 5-H), 11.33 (s, 1H OH). ^13^C NMR (176 MHz, DMSO-*d*_6_): *δ*_C_ ppm 111.0 (Pz C-4), 118.1 (Ph C-2,6), 122.3 (Pyr C-2,6), 126.7 (Ph C-4), 128.6 (C(O)CHCH), 129.6 (Ph C-3,5), 132.1 (Pz C-5), 138.4 (Pyr C-4), 138.7 (Ph C-1), 142.2 (C(O)CHCH), 150.2 (Pyr C-3,5), 161.9 (Pz C-3), 182.0 (C=O). ^15^N NMR (71 MHz, DMSO-*d*_6_): δ_N_ ppm −181.8 (Pz N-1), −118.1 (Pz N-2), −63.9 (Pyr N). HRMS (ESI^+^) for C_17_H_13_N_3_O_2_ ([M + H]^+^) calcd 292.1081, found 292.1081.

##### (2*E*)-1-(3-Hydroxy-1-phenyl-1*H*-pyrazol-4-yl)-3-(pyridin-3-yl)prop-2-en-1-one (**8i**)

The reaction mixture was stirred for 3.5 h. Yellow solid; yield 48% (1.40 mg); m.p. 230–231 °C; *R_f_* = 0.21 (DCM/MeOH 100/3, *v*/*v*). IR (*ν*_max_, cm^−1^): 3115, 3026, 1656 (C=O), 1583, 1510, 1455, 1320, 1216, 1061, 800, 748, 703, 678. ^1^H NMR (700 MHz, DMSO-*d*_6_): *δ*_H_ ppm 7.33–7.35 (m, 1H, Ph 4-H), 7.50–7.54 (m, 3H, Ph 3,5-H, Pyr 5-H), 7.70 (d, *J* = 15.8 Hz, 1H, C(O)CH), 7.81–7.84 (m, 3H, Ph 2,6-H, C(O)CHCH), 8.19–8.21 (m, 1H, Pyr 4-H), 8.61 (dd, *J* = 4.7, 1.7 Hz, 1H, Pyr 6-H), 8.97 (d, *J* = 2.3 Hz, 1H, Pyr 2-H), 9.19 (s, 1H, Pz 5-H), 11.22 (s, 1H, OH).^13^C NMR (176 MHz, DMSO-*d*_6_): *δ*_C_ ppm 111.0 (Pz C-4), 118.1 (Ph C-2,6), 124.0 (Pyr C-5), 126.1 (C(O)CH), 126.6 (Ph C-4), 129.6 (Ph C-3,5), 130.6 (Pyr C-3), 131.9 (Pz C-5), 134.8 (Pyr C-4), 137.9 (C(O)CHCH), 138.8 (Ph C-1), 149.8 (Pyr C-2), 150.7 (Pyr C-6), 161.9 (Pz C-3), 182.1 (C=O). ^15^N NMR (71 MHz, DMSO-*d*_6_): δ_N_ ppm −182.1 (Pz N-1), −119.4 (Pz N-2), −65.0 (Pyr N). HRMS (ESI^+^) for C_17_H_13_N_3_O_2_ ([M + H]^+^) calcd 292.1080, found 292.1081.

##### (2*E*)-1-(3-Hydroxy-1-phenyl-1*H*-pyrazol-4-yl)-3-(quinolin-4-yl)prop-2-en-1-one (**8j**)

The reaction mixture was stirred for 4 h. Orange solid; yield 63% (2.15 mg); m.p. 240–241 °C; *R_f_* = 0.24 (DCM/MeOH 100/3, *v*/*v*). IR (*ν*_max_, cm^−1^): 3111, 1654 (C=O), 1574, 1507, 1441, 1221, 1050, 746, 730, 684, 669. ^1^H NMR (700 MHz, DMSO-*d*_6_): *δ*_H_ ppm 7.34–7.36 (m, 1H, NPh 4-H), 7.52–7.54 (m, 2H, NPh 3,5-H), 7.74–7.76 (m, 1H, Quin 6-H), 7.86–7.87 (m, 2H, NPh 2,6-H), 7.88–7.89 (m, 1H, Quin 7-H), 7.95–7.98 (m, 2H, C(O)CHCH, Quin 3-H), 8.12–8.13 (m, 1H, Quin 5-H), 8.35–8.36 (m, 1H, Quin 8-H), 8.39 (d, *J* = 14.0 Hz, 1H, C(O)CHCH), 9.04 (d, *J* = 4.5 Hz, 1H, Quin 2-H), 9.22 (s, 1H, Pz 5-H), 11.43 (s, 1H, OH). ^13^C NMR (176 MHz, DMSO-*d*_6_): *δ*_C_ ppm 111.0 (Pz C-4), 118.1 (NPh C-2,6), 118.5 (Quin C-3), 123.8 (Quin C-8), 125.8 (Quin C-4), 126.7 (NPh C-4), 127.7 (Quin C-6), 129.1 (Quin C-5), 129.6 (NPh C-3,5), 130.2 (Quin C-7), 131.0 (C(O)CHCH), 132.2 (Pz C-4), 134.8 (C(O)CHCH), 138.7 (NPh C-1), 140.7 (Quin C-4a), 147.5 (Quin C-8a), 149.9 (Quin C-2), 161.9 (Pz C-3), 181.8 (C=O). ^15^N NMR (71 MHz, DMSO-*d*_6_): *δ*_N_ ppm −181.1 (Pz N-1). HRMS (ESI^+^) for C_21_H_15_N_3_O_2_ ([M + H]^+^) calcd 342.1237, found 342.1237.

##### (2*E*)-1-(3-Hydroxy-1-phenyl-1*H*-pyrazol-4-yl)-3-(thiophen-2-yl)prop-2-en-1-one (**8k**)

The reaction mixture was stirred for 4 h. Yellow solid; yield 45% (1.33 mg); m.p. 196–197 °C; *R_f_* = 0.54 (DCM/MeOH 100/3, *v*/*v*). IR (*ν*_max_, cm^−1^): 3104, 3067, 1647 (C=O), 1582, 1509, 1457, 1322, 1217, 1062, 967, 825, 699, 685. ^1^H NMR (700 MHz, DMSO-*d*_6_): *δ*_H_ ppm 7.17–7.18 (m, 1H, Th 5-H), 7.31–7.34 (m, 1H, Ph 4-H), 7.46 (d, *J* = 15.4 Hz, 1H, C(O)CHCH), 7.50–7.52 (m, 2H, Ph 3,5-H), 7.58–7.59 (m, 1H, Th 3-H), 7.74–7.75 (m, 1H, Th 4-H), 7.84–7.88 (m, 3H, Ph 2,6-H, C(O)CHCH), 9.08 (s, 1H, Pz 5-H), 11.31 (s, 1H, OH). ^13^C NMR (176 MHz, DMSO-*d*_6_): *δ*_C_ ppm 111.0 (Pz C-4), 118.1 (Ph C-2,6), 122.7 (C(O)CHCH), 126.5 (Ph C-4), 128.7 (Th C-5), 129.5 (Ph C-3,5), 129.8 (Th C-4), 131.6 (Pz C-5), 132.9 (Th C-3), 134.5 (C(O)CHCH), 138.8 (Ph C-1), 139.9 (Th C-2), 161.7 (Pz C-3), 182.2 (C=O). ^15^N NMR (71 MHz, DMSO-*d*_6_): *δ*_N_ ppm −181.6 (Pz N-1), −118.4 (Pz N-2). HRMS (ESI^+^) for C_16_H_12_N_2_O_2_S ([M + Na]^+^) calcd 319.0512, found 319.0512.

##### (2*E*)-3-(Furan-3-yl)-1-(3-hydroxy-1-phenyl-1*H*-pyrazol-4-yl)prop-2-en-1-one (**8l**)

The reaction mixture was stirred for 5 h. White solid; yield 95% (2.67 mg); m.p. 195–196 °C; *R_f_* = 0.62 (DCM/MeOH 100/3, *v*/*v*). IR (*ν*_max_, cm^−1^): 3111, 3071, 1653 (C=O), 1587, 1511, 1458, 1322, 1219, 1063, 1156, 727, 680. ^1^H NMR (700 MHz, DMSO-*d*_6_): *δ*_H_ ppm 6.67–6.68 (m, 1H, Furanyl 5-H), 7.00 (m, 1H, Furanyl 4-H), 7.31–7.33 (m, 1H, Ph 4-H), 7.49–7.53 (m, 4H, Ph 3,5-H, C(O)CHCH), 7.85–7.86 (m, 2H, Ph 2,6-H), 7.89 (m, 1H, Furanyl 2-H), 9.04 (s, 1H, Pz 5-H), 11.35 (s, 1H, OH). ^13^C NMR (176 MHz, DMSO-*d*_6_): *δ*_C_ ppm 111.1 (Pz C-4), 113.0 (Furanyl C-5), 116.8 (Furanyl C-4), 118.1 (Ph C-2,6), 121.1 (C(O)CHCH), 126.5 (Ph C-4), 128.1 (C(O)CHCH), 129.6 (Ph C-3,5), 131.5 (Pz C-5), 138.8 (Ph C-1), 145.9 (Furanyl C-2), 151.2 (Furanyl C-3), 161.6 (Pz C-3), 182.2 (C=O). ^15^N NMR (71 MHz, DMSO-*d*_6_): *δ*_N_ −181.6 (Pz N-1). HRMS (ESI^+^) for C_16_H_12_N_2_O_3_ ([M + Na]^+^) calcd 303.0740, found 303.0740.

#### 3.2.6. General Procedure for the Synthesis of **9a**–**i**

To a solution of an appropriate compound **8** (1 mmol) in abs. DMF (3 mL), NaH (60% suspension in mineral oil, 0.04 g, 1 mmol) and an appropriate alkylhalide (1.1 mmol) were added [76]. The reaction mixture was stirred at room temperature for 1–2 h, diluted with KHSO_4_ aq. (10 mL) and extracted with EtOAc (2 × 10 mL). The organic layers were combined, washed with H_2_O (4 × 20 mL), dried with sodium sulphate, filtrated off and concentrated. The residue was purified by column chromatography (SiO_2_, eliuent: hexane/ethyl acetate 9/1) to produce pure **9a**–**i**.

##### (2*E*)-1-(3-Methoxy-1-phenyl-1*H*-pyrazol-4-yl)-3-phenylprop-2-en-1-one (**9a**)

The reaction mixture was stirred for 1 h. Yellow solid; yield 69% (210 mg); m.p. 163–164 °C; *R_f_* = 0.78 (DCM/MeOH 100/3, *v*/*v*). IR (*ν*_max_, cm^−1^): 3118, 3089, 1652 (C=O), 1550, 1498, 1408, 1308, 1217, 756, 731, 685, 672. ^1^H NMR (700 MHz, CDCl_3_): *δ*_H_ ppm 4.16 (s, 3H, CH_3_), 7.30–7.32 (m, 1H, NPh 4-H), 7.39–7.43 (m, 3H, CPh 3,4,5-H), 7.45–7.48 (m, 2H, NPh 3,5-H), 7.63 (d, *J* = 15.7 Hz, 1H, C(O)CHCHPh), 7.64–7.66 (m, 2H, CPh 2,6-H), 7.68–7.69 (m, 2H, NPh 2,6-H), 7.82 (d, *J* = 15.7 Hz, 1H, C(O)CHCHPh), 8.42 (s, 1H, Pz 5-H). ^13^C NMR (176 MHz, CDCl_3_): *δ*_C_ ppm 56.9 (CH_3_), 112.3 (Pz C-4), 118.6 (NPh C-2,6), 124.3 (C(O)CHCHPh), 126.9 (NPh C-4), 128.5 (CPh C-2,6), 128.8 (CPh C-3,5), 129.5 (NPh C-3,5), 130.2 (CPh C-4), 131.3 (Pz C-5), 135.2 (CPh C-1), 139.2 (NPh C-1), 142.6 (C(O)CHCHPh), 162.5 (Pz C-3), 183.3 (C=O). ^15^N NMR (71 MHz, CDCl_3_): *δ*_N_ ppm −181.4 (Pz N-1), −118.5 (Pz N-2). HRMS (ESI^+^) for C_19_H_16_N_2_O_2_ ([M + Na]^+^) calcd 327.1104, found 327.1104.

##### (2*E*)-3-(4-Fluorophenyl)-1-(3-methoxy-1-phenyl-1*H*-pyrazol-4-yl)prop-2-en-1-one (**9****b**)

The reaction mixture was stirred for 2 h. Yellow solid; yield 43% (139 mg); m.p. 155–156 °C; *R_f_* = 0.53 (EtOAc/Hex 1/4, *v*/*v*). IR (*ν*_max_, cm^−1^): 3116, 3049, 1656 (C=O), 1560, 1500, 1408, 1327, 1221, 746, 706, 684, 641. ^1^H NMR (700 MHz, DMSO-*d*_6_): *δ*_H_ ppm 4.04 (s, 3H, CH_3_), 7.30–7.33 (m, 2H, CPh 3,5-H), 7.35–7.37 (m, 1H, NPh 4-H), 7.53–7.55 (m, 2H), 7.61 (d, *J* = 15.7 Hz, 1H, C(O)CHCHPh), 7.66 (d, *J* = 15.6 Hz, 1H, C(O)CHCHPh), 7.86–7.89 (m, 4H, NPh 2,6-H, CPh 2,6-H), 9.27 (s, 1H, Pz 5-H). ^13^C NMR (176 MHz, DMSO-*d*_6_): *δ*_C_ ppm 54.9 (CH_3_), 109.6 (Pz C-4), 114.3 (d, ^2^*J*= 21.7 Hz, CPh C-3,5), 116.5 (NPh C-2,6), 122.6 (C(O)CH), 125.1 (NPh C-4), 127.9 (NPh C-3,6), 129.1 (d, ^3^*J*= 8.5 Hz, CPh C-2,6), 129.7 (d, ^4^*J* = 3.1 Hz, CPh C-1), 131.4 (Pz C-5), 137.1 (NPh C-1), 138.5 (C(O)CHCH), 160.9 (Pz C-3), 161.6 (d, ^1^*J* = 248.3 Hz, CPh C-4), 179.6 (C=O). HRMS (ESI^+^) for C_19_H_15_FN_2_O_2_ ([M + H]^+^) calcd 323.1190, found 323.1191.

##### (2*E*)-3-(4-Chlorophenyl)-1-(3-methoxy-1-phenyl-1*H*-pyrazol-4-yl)prop-2-en-1-one (**9c**)

The reaction mixture was stirred for 2 h. Yellow solid; yield 80% (272 mg); m.p. 173–174 °C; *R_f_* = 0.69 (DCM/MeOH 100/1, *v*/*v*). IR (*ν*_max_, cm^−1^): 3122, 3069, 1659 (C=O), 1597, 1560, 1490, 1403, 1329, 1222, 813, 744, 683, 638 497. ^1^H NMR (700 MHz, DMSO-*d*_6_): *δ*_H_ ppm 4.04 (s, 3H, CH_3_), 7.34–7.36 (m, 1H, NPh 4-H), 7.52–7.55 (m, 4H, NPh 3,5-H, CPh 2,6-H), 7.63 (d, *J* = 15.6 Hz, 1H, C(O)CHCHPh), 7.66 (d, *J* = 15.7 Hz, 1H, C(O)CHCHPh), 7.82–7.83 (m, 2H, CPh 3,5-H), 7.88–7.89 (m, 2H, NPh 2,6-H), 9.27 (s, 1H, Pz 5-H). ^13^C NMR (176 MHz, DMSO-*d*_6_): *δ*_C_ ppm 56.5 (CH_3_), 111.2 (Pz C-4), 118.1 (NPh C-2,6), 125.1 (C(O)CHCHPh), 126.7 (NPh C-4), 129.0 (CPh C-2,6), 129.6 (NPh C-3,5), 130.2 (CPh C-3,5), 133.1(Pz C-5), 133.7 (CPh C-1), 134.8 (CPh C-4), 138.7 (NPh C-4), 140.0 (C(O)CHCHPh), 162.6 (Pz C-3), 181.2 (C=O). ^15^N NMR (71 MHz, DMSO-*d*_6_): *δ*_N_ ppm −182.2 (Pz N-1), −119.1 (Pz N-2). HRMS (ESI^+^) for C_19_H_15_ClN_2_O_2_ ([M + Na]^+^) calcd 361.0714, found 361.0714.

##### (2*E*)-3-[4-(Dimethylamino)phenyl]-1-(3-methoxy-1-phenyl-1*H*-pyrazol-4-yl)prop-2-en-1-one (**9d**)

The reaction mixture was stirred for 1 h. Yellow solid; yield 43% (277 mg); m.p. 186–187 °C; *R_f_*=0.15 (EtOAc/Hex 1/4, *v*/*v*). IR (*ν*_max_, cm^−1^): 3111, 1634 (C=O), 1580, 1503, 1421, 1358, 1157, 1032, 810, 748, 688, 667. ^1^H NMR (700 MHz, DMSO-*d*_6_): *δ*_H_ ppm 3.00 (s, 6H, N(CH_3_)_2_), 4.04 (s, 3H, OCH_3_), 6.75–6.76 (m, 2H, CPh 3,5-H), 7.33–7.35 (m, 1H, NPh 4-H), 7.41 (d, *J* = 15.4 Hz 1H C(O)CHCHPh), 7.51–7.54 (m, 2H), 7.57–7.61 (m, 3H, CPh 2,6-H, C(O)CHCHPh), 7.88–7.90 (m, 2H, NPh 2,6-H), 9.16 (s, 1H, Pz 5-H). ^13^C NMR (176 MHz, DMSO-*d*_6_): *δ*_C_ ppm 39.7 (N(CH_3_)_2_), 56.5 (OCH_3_), 111.7 (Pz C-4), 111.8 (CPh 3,5-C), 118.0 (NPh C-2,6), 118.8 (C(O)CHCHPh), 122.0 (CPh C-1), 126.5 (NPh C-4), 129.5 (CPh C-2,6), 130.2 (NPh C-3,5), 132.4 (Pz C-5), 138.8 (NPh C-1), 142.5 (C(O)CHCHPh), 151.8 (CPh C-4), 162.3 (Pz C-3), 181.3 (C=O). HRMS (ESI^+^) for C_21_H_21_N_3_NaO_2_ ([M + Na]^+^) calcd 370.1526, found 370.1526.

##### (2*E*)-1-(3-Methoxy-1-phenyl-1*H*-pyrazol-4-yl)-3-(quinolin-4-yl)prop-2-en-1-one (**9e**)

The reaction mixture was stirred for 1 h. Yellow solid; yield 72% (256 mg); m.p. 182–183 °C, *R_f_* = 0.6 (DCM/MeOH 100/3, *v*/*v*). IR (*ν*_max_, cm^−1^): 3120, 3091, 1654 (C=O), 1596, 1556, 1500, 1414, 1309, 1218, 834, 749, 726, 682. ^1^H NMR (700 MHz, DMSO-*d*_6_): *δ*_H_ ppm 4.06 (s, 3H, CH_3_), 7.35–7.38 (m, 1H, NPh 4-H), 7.53–7.56 (m, 2H, NPh 3,5-H), 7.71–7.73 (m, 1H, Quin 7-H), 7.83–7.85 (m, 1H, Quin 6-H), 7.87–7.91 (m, 3H, C(O)CHCH, NPh 2,6-H), 7.95–7.96 (m, 1H, Quin 4-H), 8.10–8.11 (m, 1H, Quin 5-H), 8.31–8.32 (m, 1H, Quin 8-H), 8.36 (d, *J* = 15.5 Hz, 1H, C(O)CHCH), 9.01 (m, 1H, Quin 2-H), 9.34 (s, 1H, Pz 5-H). ^13^C NMR (176 MHz, DMSO-*d*_6_): *δ*_C_ ppm 56.6 (CH_3_), 111.1 (Pz C-4), 118.2 (NPh C-2,6), 118.5 (Quin C-3), 123.6, 125.7, 126.9, 127.5, 129.6 (NPh C-3,5), 129.7, 129.8, 130.7, 133.4 (Pz C-5), 135.1 (C(O)CHCH), 138.7 (NPh C-1), 139.8 (Quin C-4a), 148.3 (Quin C-8a), 150.3 (Quin C-2), 162.7 (Pz C-3), 181.0 (C=O). HRMS (ESI^+^) for C_22_H_17_N_3_O_2_ ([M + H]^+^) calcd 356.1394, found 356.1394.

##### (2*E*)-1-(3-Methoxy-1-phenyl-1*H*-pyrazol-4-yl)-3-(thiophen-2-yl)prop-2-en-1-one (**9f**)

The reaction mixture was stirred for 1 h. Yellow solid; yield 60% (186 mg); m.p. 131–132 °C; *R_f_* = 0.37 (EtOAc/Hex 1/4, *v*/*v*). IR (*ν*_max_, cm^−1^): 3127, 3096, 1664 (C=O), 1560, 1502, 1410, 1399, 1225, 1014, 943, 750, 685, 669, 505. ^1^H NMR (700 MHz, DMSO-*d*_6_): *δ*_H_ ppm 4.05 (s, 3H, CH_3_), 7.18–7.19 (m, 1H, Th 5-H), 7.34–7.38 (m, 2H, Ph 4-H, C(O)CHCH), 7.51–7.54 (m, 2H, Ph 3,5-H), 7.60 (m, 1H, Th 3-H), 7.75–7.76 (m, 1H, Th 4-H), 7.84 (d, *J* = 15.3 Hz, 1H, C(O)CHCH), 7.88–7.91 (m, 2H, Ph 2,6-H), 9.21 (s, 1H, Pz 5-H). ^13^C NMR (176 MHz, DMSO-*d*_6_): *δ*_H_ ppm 57.1 (CH_3_), 111.6 (Pz C-4), 118.6 (Ph C-2,6), 123.3 (C(O)CHCH), 127.1 (Ph C-4), 129.1 (Th C-5), 130.0 (Ph C-3,5), 130.2 (Th C-4), 133.2 (Pz C-5), 133.3 (Th C-3), 135.0 (C(O)CHCH), 139.2 (Ph C-1), 140.3 (Th C-2), 162.8 (Pz C-3), 181.4 (C=O). ^15^N NMR (71 MHz, DMSO-*d*_6_): *δ*_N_ ppm −181.9 (Pz N-1), −119.4 (Pz N-2). HRMS (ESI^+^) for C_17_H_14_N_2_O_2_S ([M + Na]^+^) calcd 333.0668, found 333.0668.

##### (2*E*)-3-(Furan-3-yl)-1-(3-methoxy-1-phenyl-1*H*-pyrazol-4-yl)prop-2-en-1-one (**9g**)

The reaction mixture was stirred for 1 h. Yellow solid; yield 77% (227 mg); m.p. 136–137 °C; *R_f_* = 0.34 (EtOAc/Hex 1/4, *v*/*v*). IR (*ν*_max_, cm^−1^): 3122, 3063, 1658 (C=O), 1558, 1501, 1461, 1404, 1220, 1014, 742, 681, 631, 595. ^1^H NMR (700 MHz, DMSO-*d*_6_): *δ*_H_ ppm 4.05 (s, 3H, CH_3_), 6.67–6.68 (m, 1H, Furanyl 4-H), 7.01 (m, 1H, Furanyl 5-H), 7.33–7.35 (m, 1H, Ph 4-H), 7.40 (d, *J* = 14.0 Hz, 1H, C(O)CHCH), 7.48–7.53 (m, 3H, Ph 3,5-H, C(O)CHCH), 7.90–7.91 (m, 3H, Ph 2,6-H, Furanyl 2-H), 9.16 (s, 1H, Pz 5-H). ^13^C NMR (176 MHz, DMSO-*d*_6_): *δ*_C_ ppm 56.6 (CH_3_), 111.3 (Pz C-4), 113.0 (Furanyl C-4), 116.8 (Furanyl C-5), 118.1 (Ph C-2,6), 121.2 (C(O)CHCH), 126.7 (Ph C-4), 128.3 (C(O)CHCH), 129.5 (Ph C-3,5), 132.7 (Pz C-5), 138.7 (Ph C-1), 145.9 (Furanyl C-2), 151.1 (Furanyl C-3), 162.2 (Pz C-3), 181.1 (C=O). ^15^N NMR (71 MHz, DMSO-*d*_6_): *δ*_N_ ppm −181.6 (Pz N-1), −119.0 (Pz N-2). HRMS (ESI^+^) for C_17_H_14_N_2_O_3_ ([M + Na]^+^) calcd 317.0897, found 317.0897.

##### (2*E*)-3-Phenyl-1-(1-phenyl-3-propoxy-1*H*-pyrazol-4-yl)prop-2-en-1-one (**9h**)

The reaction mixture was stirred for 1 h. Yellow solid; yield 96% (320 mg); m.p. 131–132 °C; *R_f_* = 0.77 (DCM/MeOH 100/1, *v*/*v*). IR (*ν*_max_, cm^−1^): 3118, 3082, 1657 (C=O), 1597, 1561, 1491, 1447, 1349, 1330, 1222, 761, 746, 681, 638. ^1^H NMR (400 MHz, DMSO-*d*_6_): *δ*_H_ ppm 1.05 (t, *J* = 7.4 Hz, 3H, CH_3_), 1.85 (sext, *J* = 7.1 Hz, 2H, CH_3_CH_2_CH_2_), 4.33 (t, *J* = 6.5 Hz, 2H, CH_3_CH_2_CH_2_), 7.32–7.36 (m, 1H, NPh 4-H), 7.44–7.54 (m, 5H, NPh 3,5-H, CPh 3,4,5-H), 7.64–7.72 (m, 2H, C(O)CHCHPh), 7.74–7.78 (m, 2H, CPh 2,6-H), 7.86–7.90 (m, 2H, NPh 2,6-H), 9.20 (s, 1H, Pz 5-H). ^13^C NMR (101 MHz, DMSO-*d*_6_): *δ*_C_ ppm 10.4 (CH_3_), 22.0 (CH_3_CH_2_CH_2_), 70.6 (CH_3_CH_2_CH_2_), 111.5 (Pz C-4), 118.1 (NPh C-2,6), 124.5 (C(O)CHCHPh), 126.7 (NPh C-4), 128.3 (CPh C-2,6), 129.0 (NPh C-3,5), 129.5 (CPh C-3,5), 130.4 (CPh C-4), 132.6 (Pz C-5), 134.7 (CPh C-1), 138.7 (NPh C-1), 141.3 (C(O)CHCHPh), 161.9 (Pz C-3), 181.5 (C=O). HRMS (ESI^+^) for C_21_H_20_N_2_O_2_ ([M + Na]^+^) calcd 355.1417, found 355.1417.

##### (2*E*)-1-[3-(2-Methoxyethoxy)-1-phenyl-1*H*-pyrazol-4-yl]-3-phenylprop-2-en-1-one (**9i**)

The reaction mixture was stirred for 1 h. Yellow solid; yield 50% (175 mg); m.p. 133–134 °C; *R_f_* = 0.43 (DCM/MeOH 100/1, *v*/*v*). IR (*ν*_max_, cm^−1^): 3114, 3089, 1656 (C=O), 1596, 1556, 1493, 1468, 1371, 1220, 766, 750, 686, 677. ^1^H NMR (400 MHz, DMSO-*d*_6_): *δ*_H_ ppm 3.39 (s, 3H, CH_3_), 3.78–3.80 (m, 2H, CH_3_OCH_2_CH_2_), 4.49–4.51 (m, 2H, CH_3_OCH_2_CH_2_), 7.33–7.37 (m, 1H, NPh 4-H), 7.46–7.55 (m, 5H, NPh 3,5-H, CPh 3,4,5-H), 7.66 (d, *J* = 15.6 Hz, 1H, C(O)CHCHPh), 7.75–7.79 (m, 3H, C(O)CHCHPh, CPh 2,6-H), 7.88–7.90 (m, 2H, NPh 2,6-H), 9.19 (s, 1H, Pz 5-H). ^13^C NMR (101 MHz, DMSO-*d*_6_): *δ*_C_ ppm 58.3 (CH_3_), 68.5 (CH_3_OCH_2_CH_2_), 70.2 (CH_3_OCH_2_CH_2_), 111.5 (Pz C-4), 118.2 (NPh C-2,6), 124.5 (C(O)CHCHPh), 126.7 (NPh C-4), 128.4 (CPh C-2,6), 129.0 (CPh C-3,5), 129.5 (NPh C-3,5), 130.4 (CPh C-4), 132.7 (Pz C-5), 134.7 (CPh C-1), 138.7 (NPh C-1), 141.3 (C(O)CHCHPh), 161.7 (Pz C-3), 181.5 (C=O). HRMS (ESI^+^) for C_21_H_20_N_2_O_3_ ([M + Na]^+^) calcd 371.1366, found 371.1366.

#### 3.2.7. General Procedure for the Preparation of 3-pyridinyl- and 4-pyridinyl-1-phenyl-1*H*-pyrazol-4-ylethanones (**11a,b**)

To the solution of triflate **10** (320 mg, 1 mmol) in abs. EtOH (2.5 mL), 4-pyridinyl- or 3-pyridinylboronic acid (123 mg, 1 mmol), Cs_2_CO_3_ (652 mg, 2 mmol), KBr (36 mg, 0.3 mmol) and Pd(PPh_3_)_4_ (116 mg, 0.1 mmol) were added, and the reaction mixture was irradiated (150 W) at 80 °C temperature for 10 min. EtOH was evaporated, the mixture was diluted with water (10 mL) and extracted with ethyl acetate (3 × 10 mL). The organic layers were combined, washed with brine, dried over Na_2_SO_4_, filtrated off, and the solvent was evaporated. The residue was purified by flash column chromatography (SiO_2_, eluent: ethyl acetate/*n*-hexane, 1:4, *v*/*v*) to provide the desired compounds **11a**,**b**.

##### 1-[1-Phenyl-3-(pyridin-3-yl)-1*H*-pyrazol-4-yl]ethan-1-one (**11a**)

White solid; yield 63% (166 mg); m.p. 121–122 °C; *R_f_* = 0.36 (EtOAc, *v*/*v*). IR (*ν*_max_, cm^−1^): 3120, 3041, 1684 (C=O), 1599, 1521, 1448, 1362, 1260, 1241, 1221, 977, 940, 863, 751, 705, 683. ^1^H NMR (700 MHz, CDCl_3_): *δ*_H_ ppm 2.50 (s, 3H, CH_3_), 7.37–7.39 (m, 1H, Pyr 5-H), 7.39–7.42 (m, 1H, Ph 4-H), 7.51–7.54 (m, 2H, Ph 3,5-H), 7.78–7.79 (m, 2H, Ph 2,6-H), 8.18 (dt, *J*_4-H,5-H_ = 7.9 Hz, *J* = 2.0 Hz, 1H, Pyr 4-H), 8.48 (s, 1H, Pz 5-H), 8.65 (dd, *J*_5-H,6-H_ = 4.9 Hz, *J* = 1.7 Hz, 1H, Pyr 4-H), 9.03 (d, *J* = 2.3 Hz, 1H, Pyr 2-H). ^13^C NMR (176 MHz, CDCl_3_): *δ*_C_ ppm 29.3 (CH_3_), 119.8 (Ph C-2,6), 122.8 (Pz C-4), 122.9 (Pyr C-5), 128.1 (Ph C-4), 128.6 (Pyr C-3), 129.9 (Ph C-3,5), 132.2 (Pz C-5), 137.1 (Pyr C-4), 139.1 (NPh C-1), 149.9 (Pyr C-6), 150.2 (Pyr C-2), 150.7 (Pz C-3), 191.8 (C=O). ^15^N NMR (71 MHz, CDCl_3_): *δ*_N_ ppm −162.6 (Pz N-1), −71.4 (Pyr N). HRMS (ESI^+^) for C_16_H_13_N_3_O ([M + H]^+^) calcd 264.1131, found 264.1131.

##### 1-[1-Phenyl-3-(pyridin-4-yl)-1*H*-pyrazol-4-yl]ethan-1-one (**11b**)

White solid; yield 64% (169 g); m.p. 120–122 °C; *R_f_* = 0.16 (EtOAc/Hex 1/1, *v*/*v*). IR (*ν*_max_, cm^−1^): 3120, 3041, 1684 (C=O), 1599, 1521, 1448, 1362, 1260, 1241, 1221, 977, 940, 863, 751, 705, 683. ^1^H NMR (700 MHz, CDCl_3_): *δ*_H_ ppm 2.51 (s, 3H, CH_3_), 7.39–7.42 (m, 1H, Ph 4-H), 7.51–7.53 (m, 2H, Ph 3,5-H), 7.76–7.78 (m, 2H, Ph 2,6-H), 7.79–7.80 (m, 2H, Pyr 3,5-H), 8.47 (s, 1H, Pz 5-H), 8.69 (d, *J*_3,5-H,2,6-H_ = 6.2 Hz, 2H, Pyr 2,6-H). ^13^C NMR (176 MHz, CDCl_3_): *δ*_C_ ppm 29.5 (CH_3_), 119.9 (Ph C-2,6), 122.9 (Pz C-4), 123.9 (Pyr C-3,5), 128.3 (Ph C-4), 129.9 (Ph C-3,5), 132.5 (Pz C-5), 139.1 (Ph C-1), 140.2 (Pyr C-4), 149.9 (Pyr C-2,6), 150.9 (Pz C-3), 191.8 (C=O). ^15^N NMR (71 MHz, CDCl_3_): *δ*_N_ ppm −162.2 (Pz N-1), −70.1 (Pyr N). HRMS (ESI^+^) for C_16_H_14_N_3_O ([M + H]^+^) calcd 264.1131, found 264.1131.

#### 3.2.8. General Procedure for the Synthesis of **12a**–**j**

To a solution of an appropriate pyrazole ethanone (**11a,b**) (50 mg, 0.19 mmol) in EtOH (96%, 2 mL), NaOH (75 mg, 1.9 mmol) and appropriate benzaldehyde (0.47 mmol) were added. The reaction mixture was heated at 55 °C for 10 min. After the completion of the reaction as monitored by TLC, EtOH was evaporated, the mixture was diluted with water (10 mL) and extracted with ethyl acetate (3 × 10 mL). The organic layers were combined, washed with brine, dried over sodium sulphate, filtrated off, and the solvent was evaporated. The residue was purified by flash column chromatography (SiO_2_, eluent: ethyl acetate/*n*-hexane, 1:2, *v*/*v*) to provide the desired compounds **12a**–**j**.

##### (2*E*/*Z*)-3-Phenyl-1-[1-phenyl-3-(pyridin-3-yl)-1*H*-pyrazol-4-yl]prop-2-en-1-one (**12a**,**13a**)

A mixture of isomers was obtained in the ratio ***E*-12a**:***Z*-13a** = 1:0.15. Yellow solid; yield 70% (55 mg); m.p. 169–170.8 °C; *R_f_* = 0.67 (EtOAc/Hex 1/2, *v*/*v*). IR (*ν*_max_, cm^−1^): 3120, 3041, 1684 (C=O), 1599, 1521, 1448, 1362, 1260, 1241, 1221, 977, 940, 863, 751, 705, 683. ^1^H NMR (700 MHz, CDCl_3_): *δ*_H_ ppm 6.45 (d, *J* = 12.7 Hz, 1H, C(O)CHCHPh of minor isomer), 6.91 (d, *J* =12.7 Hz, 1H, C(O)CHCHPh of minor isomer), 7.12 (d, *J* =15.6 Hz, 1H, C(O)CHCHPh of major isomer), 7.36–7.42 (m, 5H, NPh 4-H, CPh 3,4,5-H, Pyr 5-H), 7.49–7.50 (m, 2H, CPh 2,6-H), 7.52–7.55 (m, 2H, NPh 3,5-H), 7.64–7.66 (m, 2H, NPh 2,6-H of minor isomer), 7.76 (d, *J* = 15.6 Hz, 1H, C(O)CHCHPh of major isomer), 7.82–7.83 (m, 2H, NPh 2,6-H), 8.18 (dd, *J*_Pyr 4-H,5-H_ = 7.8 Hz, *J*_Pyr 4-H,6-H_ = 1.8 Hz, 1H, Pyr 4-H), 8.32 (s, 1H, Pz 5-H of minor isomer), 8.59 (s, 1H, Pz 5-H of major isomer), 8.65 (d, *J* = 4.8 Hz, 1H, Pyr 4-H of minor isomer), 8.68 (d, *J*_Pyr 5-H,6-H_ = 4.8 Hz, 1H, Pyr 6-H of major isomer), 9.04 (s, 1H, Pyr 2-H of minor isomer), 9.07 (s, 1H, Pyr 2-H of major isomer). ^1^^3^C NMR (176 MHz, CDCl_3_): *δ*_C_ ppm 119.8 (NPh C-2,6), 123.0 (Pyr C-5), 123.5 (Pz C-4), 124.4 (C(O)CH=CHPh), 128.1 (NPh C-4), 128.5 (CPh C-2,6), 128.7 (Pyr C-3), 129.1 (CPh C-3,5), 129.9 (NPh C-3,5), 130.8 (CPh C-4), 131.5 (Pz C-5), 134.7 (CPh C-1), 137.1 (Pyr C-4), 139.2 (NPh C-1), 144.2 (C(O)CH=CHPh), 149.9 (Pyr C-6), 150.2 (Pyr C-2), 150.9 (Pz C-3), 184.4 (CO). ^15^N NMR (71 MHz, CDCl_3_): *δ*_N_ ppm −162.0 (Pz N-1), −71.4 (Pyr N). HRMS (ESI^+^) for C_23_H_18_N_3_O ([M + H]^+^) calcd 352.1444, found 352.1444.

##### (2*E*/*Z*)-3-(4-Methylphenyl)-1-[1-phenyl-3-(pyridin-3-yl)-1*H*-pyrazol-4-yl]prop-2-en-1-one (**12b**,**13b**)

A mixture of isomers was obtained in the ratio ***E*-12b**:***Z*-13b** = 1:0.17. Yellow crystals; yield 61% (42 mg); m.p. 150.2–151.3 °C; *R_f_*=0.61 (EtOAc/Hex 1/2, *v*/*v*). IR (*ν*_max_, cm^−1^): 3120, 3041, 1684 (C=O), 1599, 1521, 1448, 1362, 1260, 1241, 1221, 1231, 977, 940, 863, 751, 705, 683. ^1^H NMR (700 MHz, CDCl_3_): *δ*_H_ ppm 2.29 (s, 3H, CH_3_ of minor isomer), 2.38 (s, 3H, CH_3_ of major isomer), 6.40 (d, *J* =12.7 Hz, 1H, C(O)CHCHPh of minor isomer), 6.86 (d, *J* =12.7 Hz, 1H, C(O)CHCHPh of minor isomer), 7.07 (d, *J* =15.6 Hz, 1H, C(O)CHCHPh of major isomer), 7.19 (d, *J* = 7.8 Hz, 2H, CPh 3,5-H), 7.36–7.42 (m, 4H, NPh 4-H, CPh 2,6-H, Pyr 5-H), 7.47–7.49 (m, 2H, NPh 3,5-H of minor isomer), 7.51–7.54 (m, 2H, NPh 3,5-H of major isomer), 7.65–7.67 (m, 2H, NPh 2,6-H of minor isomer), 7.74 (d, *J* = 15.6 Hz, 1H, C(O)CHCHPh of major isomer), 7.81–7.83 (m, 2H, NPh 2,6-H), 8.17 (dt, *J*_Pyr 4-H,5-H_ = 7.9 Hz, *J* = 2.0 Hz, 1H, Pyr 4-H), 8.34 (s, 1H, Pz 5-H of minor isomer), 8.57 (s, 1H, Pz 5-H, of major isomer), 8.67 (dd, *J*_Pyr 5-H,6-H_ = 4.9 Hz, *J* = 1.7 Hz 1H, Pyr 4-H), 9.04 (d, *J* = 2.2 Hz, 1H, Pyr 2-H of minor isomer), 9.07 (d, *J* = 2.0 Hz, 1H, Pyr 2-H of major isomer). ^1^^3^C NMR (176 MHz, CDCl_3_): *δ*_C_ ppm 21.7 (CH_3_), 119.8 (NPh C-2,6), 123.0 (Pyr C-5), 123.4 (Pz C-4), 123.6 (C(O)CHCHPh), 128.0 (NPh C-4), 128.6 (CPh C-2,6), 128.8 (Pyr C-3), 129.2 129.9 (CPh C-3,5, NPh C-3,5), 131.4 (Pz C-5), 131.9 (CPh C-1), 137.1 (Pyr C-4), 139.2 (NPh C-1), 141.4 (CPh C-4), 144.3 (C(O)CHCHPh), 149.9 (Pyr C-6), 150.2 (Pyr C-2), 150.8 (Pz C-3), 184.6 (C(O)CHCHPh). ^15^N NMR (71 MHz, CDCl_3_): *δ*_N_ ppm −161.9 (Pz N-1), −70.8 (Pyr N). HRMS (ESI^+^) for C_24_H_20_N_3_O ([M + H]^+^) calcd 366.1601, found 366.1601.

##### (2*E*/*Z*)-1-[1-Phenyl-3-(pyridin-3-yl)-1*H*-pyrazol-4-yl]-3-[4-(trifluoromethoxy)phenyl]prop-2-en-1-one (**12c**,**13c**)

A mixture of isomers was obtained in the ratio ***E*-12c**:***Z*-13c** = 1:0.4. Yellow crystals; yield 55% (46 mg); m.p. 183–184 °C; *R_f_* = 0.57 (EtOAc/Hex 1/2, *v*/*v*). IR (*ν*_max_, cm^−1^): 3120, 3041, 1684 (C=O), 1599, 1521, 1448, 1362, 1260, 1241, 1221, 1165, 977, 940, 863, 751, 705, 683. ^1^H NMR (700 MHz, CDCl_3_): *δ*_H_ ppm 6.52 (d, *J* =12.8 Hz, 1H, C(O)CHCHPh of minor isomer), 6.84 (d, *J* =12.8 Hz, 1H, C(O)CHCHPh of minor isomer), 7.06 (d, *J* =15.6 Hz, 1H, C(O)CHCHPh of major isomer), 7.12–7.13 (m, 2H, CPh 3,5-H of minor isomer), 7.22–7.23 (m, 2H, CPh 3,5-H of major isomer), 7.37–7.43 (m, 2H, NPh 4-H, Pyr 5-H), 7.49–7.55 (m, 4H, CPh 2,6-H, NPh 3,5-H), 7.73 (d, *J* = 15.6 Hz, 1H, C(O)CHCHPh of major isomer), 7.81–7.83 (m, 2H, NPh 2,6-H), 8.17 (d, *J*_Pyr 4-H,5-H_ = 7.8 Hz, 1H, Pyr 4-H), 8.41 (s, 1H, Pz 5-H of minor isomer), 8.59 (s, 1H, Pz 5-H of major isomer), 8.66 (d, *J*_Pyr 5-H,6-H_ = 4.7 Hz, 1H, Pyr 6-H of major isomer), 8.69 (d, *J*_Pyr 5-H,6-H_ = 4.3 Hz, 1H, Pyr 6-H of minor isomer), 9.03 (s, 1H, Pyr 2-H of minor isomer), 9.06 (s, 1H, Pyr 2-H of major isomer). ^13^C NMR (176 MHz, CDCl_3_): *δ*_C_ ppm 119.8 (NPh C-2,6), 121.3 (CPh C-3,5), 123.1 (Pyr C-5), 123.4 (Pz C-4), 125.1 (C(O)CHCHPh), 127.9 (NPh C-4 of minor isomer), 128.2 (NPh C-4 of major isomer), 128.7 (Pyr C-3), 129.9 (CPh C-2,6, NPh C-3,5), 130.5 (q, ^1^*J* = 254.6 Hz, OCF_3_), 131.6 (Pz C-5 of major isomer), 133.3 (CPh C-1 of major isomer), 133.6 (CPh C-1 of minor isomer), 137.0 (Pyr C-4 of minor isomer), 137.1 (Pyr C-4 of major isomer), 138.6 (NPh C-1 of minor isomer), 139.1 (NPh C-1 of major isomer), 142.3 (C(O)CHCHPh), 150.0 (Pyr C-6), 150.2 (Pyr C-2), 150.7 (Pz C-3), 150.9 (CPh C-4), 184.0 (C(O)CHCHPh of major isomer), 186.8 (C(O)CHCHPh of minor isomer). ^15^N NMR (71 MHz, CDCl_3_): *δ*_N_ ppm −161.6 (Pz N-1), −70.9 (Pyr N). HRMS (ESI^+^) for C_24_H_17_N_3_O_2_F_3_ ([M + H]^+^) calcd 436.1267, found 436.1267.

##### (2*E*/*Z*)-3-Phenyl-1-[1-phenyl-3-(pyridin-4-yl)-1*H*-pyrazol-4-yl]prop-2-en-1-one (**12d**,**13d**)

A mixture of isomers was obtained in the ratio ***E*-12d**:***Z*-13d** = 1:0.18. White crystals; yield 58% (39 mg); m.p. 162.8–165 °C; *R_f_* = 0.29 (EtOAc/Hex 1/4, *v*/*v*). IR (*ν*_max_, cm^−1^): 3120, 3041, 1684 (C=O), 1599, 1521, 1448, 1362, 1260, 1241, 1221, 977, 940, 863, 751, 705, 683. ^1^H NMR (700 MHz, CDCl_3_): *δ*_H_ ppm 6.47 (d, *J* =12.7 Hz, 1H, C(O)CHCHPh of minor isomer), 6.93 (d, *J* =12.7 Hz, 1H, C(O)CHCHPh of minor isomer), 7.12 (d, *J* =15.7 Hz, 1H, C(O)CHCHPh of major isomer), 7.38–7.42 (m, 4H, NPh 4-H, CPh 3,4,5-H), 7.48–7.51 (m, 2H, CPh 2,6-H), 7.52–7.54 (m, 2H, NPh 3,5-H), 7.63–7.64 (m, 2H, NPh 2,6-H of minor isomer), 7.76 (d, *J* = 15.7 Hz, 1H, C(O)CHCHPh of major isomer), 7.79–7.83 (m, 4H, Pyr 3,5-H, NPh 2,6-H), 8.31 (s, 1H, Pz 5-H of minor isomer), 8.56 (s, 1H, Pz 5-H of major isomer), 8.68–8.71 (m, 2H, Pyr 2,6-H). ^1^^3^C NMR (176 MHz, CDCl_3_): *δ*_C_ ppm 119.8 (NPh C-2,6), 123.6 (Pz C-4), 123.8 (Pyr C-3,5), 124.5 (C(O)CHCHPh), 128.2 (NPh C-4), 128.6 (CPh C-2,6), 129.1 (CPh C-3,5), 129.9 (NPh C-3,5), 130.9 (CPh C-4), 131.7 (Pz C-5), 134.5 (CPh C-1), 139.1 (NPh C-1), 140.3 (Pyr C-4), 144.4 (C(O)CHCHPh), 149.9 (Pyr C-2,6), 151.1 (Pz C-3), 184.6 (C(O)CHCHPh). ^15^N NMR (71 MHz, CDCl_3_): *δ*_N_ ppm −161.5 (Pz N-1), −70.6 (Pyr N). HRMS (ESI^+^) for C_23_H_18_N_3_O ([M + H]^+^) calcd 352.1444, found 352.1444.

##### (2*E*/*Z*)-3-(4-Methylphenyl)-1-[1-phenyl-3-(pyridine-4-yl)-1*H*-pyrazol-4-yl]prop-2-en-1-one (**12e**,**13e**)

A mixture of isomers was obtained in the ratio ***E*-12e**:***Z*-13e** = 1:0.13. Yellow crystals; yield 58% (40 mg); m.p. 176–177 °C; *R_f_*=0.35 (EtOAc/Hex 1/2, *v*/*v*). IR (*ν*_max_, cm^−1^): 3120, 3041, 1684 (C=O), 1599, 1521, 1448, 1362, 1260, 1241, 1221, 977, 940, 863, 751, 705, 683. ^1^H NMR (700 MHz, CDCl_3_): *δ*_H_ ppm 2.29 (s, 3H, CH_3_ of minor isomer), 2.38 (s, 3H, CH_3_ of major isomer), 6.42 (d, *J* =12.7 Hz, 1H, C(O)CHCHPh of minor isomer), 6.89 (d, *J* =12.7 Hz, 1H, C(O)CH=CHPh of minor isomer), 7.07 (d, *J* =15.6 Hz, 1H, C(O)CHCHPh of major isomer), 7.19 (d, *J* = 7.8 Hz, 2H, CPh 3,5-H), 7.38–7.42 (m, 3H, NPh 4-H, CPh 2,6-H), 7.47–7.50 (m, 2H, NPh 3,5-H of minor isomer), 7.52–7.54 (m, 2H, NPh 3,5-H of major isomers), 7.64–7.66 (m, 2H, NPh 2,6-H of minor isomer), 7.74 (d, *J* = 15.6 Hz, 1H, C(O)CHCHPh of major isomer), 7.79–7.83 (m, 4H, Pyr 3,5-H, NPh 2,6-H), 8.32 (s, 1H, Pz 5-H of minor isomer), 8.55 (s, 1H, Pz 5-H of major isomer), 8.70 (d, ^3^*J* = 5.3 Hz, 1H, Pyr 2,6-H). ^1^^3^C NMR (176 MHz, CDCl_3_): *δ*_C_ ppm 21.7 (CH_3_), 119.9 (NPh C-2,6), 123.6 (C(O)CHCHPh), 123.7 (Pz C-4), 123.8 (Pyr C-3,5), 128.1 (NPh C-4), 128.6 (CPh C-2,6), 128.8 (Pyr C-1), 129.9 (CPh C-3,5, NPh C-3,5), 131.6 (Pz C-5), 131.8 (CPh C-1), 139.1 (NPh C-1), 141.5 (CPh C-4), 144.6 (C(O)CHCHPh), 149.9 (Pyr C-2,6), 151.0 (Pz C-3), 184.8 (C(O)CHCHPh). ^15^N NMR (71 MHz, CDCl_3_): *δ*_N_ ppm −161.7 (Pz N-1), −70.2 (Pyr N). HRMS (ESI^+^) for C_24_H_20_N_3_O ([M + H]^+^) calcd 366.1601, found 366.1601.

##### (2*E*/*Z*)-1-[1-Phenyl-3-(pyridine-4-yl)-1*H*-pyrazol-4-yl]-3-[4-(trifluoromethoxy)phenyl]prop-2-en-1-one (**12f**,**13f**)

A mixture of isomers was obtained in ratio ***E*-12f**:***Z*-13f** = 1:0.24. Yellow crystals; yield 51% (42 mg); m.p. 168–170 °C; *R_f_* = 0.42 (EtOAc/Hex 1/2, *v*/*v*). IR (*ν*_max_, cm^−1^): 3120, 3041, 1684 (C=O), 1599, 1521, 1448, 1362, 1260, 1241, 1221, 1165, 977, 940, 863, 751, 705, 683. ^1^H NMR (700 MHz, CDCl_3_): *δ*_H_ ppm 6.54 (d, *J* =12.8 Hz, 1H, C(O)CHCHPh of minor isomer), 6.87 (d, *J* =12.8 Hz, 1H, C(O)CHCHPh of minor isomer), 7.07 (d, *J* =15.6 Hz, 1H, C(O)CHCHPh of major isomer), 7.12–7.13 (m, 2H, CPh 3,5-H of minor isomer), 7.20–7.24 (m, 2H, CPh 3,5-H of major isomer), 7.40–7.43 (m, 1H, NPh 4-H), 7.49–7.56 (m, 4H, CPh 2,6-H, NPh 3,5-H), 7.68–7.69 (m, 2H, NPh 2,6-H of minor isomer), 7.73 (d, *J* = 15.7 Hz, 1H, C(O)CHCHPh of major isomer), 7.79–7.82 (m, 4H, NPh 2,6-H, Pyr 3,5-H), 8.57 (s, 1H, Pz 5-H), 8.67–8.77 (m, 2H, Pyr 2,6-H). ^13^C NMR (176 MHz, CDCl_3_): *δ*_C_ ppm 119.8 (NPh C-2,6 of minor isomer), 119.9 (NPh C-2,6 of major isomer), 120.7 (CPh C-3,5 of minor isomer), 121.4 (CPh C-3,5 of major isomers), 123.5 (Pyr C-3,5), 123.9 (Pz C-4), 125.2 (C(O)CH=CHPh), 127.9 (NPh C-4 of minor isomer), 128.3 (NPh C-4 of major isomer), 128.4 (Pyr C-4), 129.9 (NPh C-3,5 of major isomer), 130.0 (CPh C-2,6), 130.7 (q, ^1^*J* = 266.6 Hz, OCF_3_), 131.7 (Pz C-5), 133.1 (CPh C-1), 138.9 (NPh C-1 of minor isomer), 139.1 (NPh C-1 of major isomer), 140.3 (Pyr C-4 of major isomer), 140.7 (Pyr C-4 of minor isomer), 142.5 (C(O)CHCHPh), 149.8 (Pyr C-2,6 of minor isomer), 149.9 (Pyr C-2,6 of major isomer), 150.8 (CPh C-4 of major isomer), 151.0 (CPh C-4 of minor isomer), 151.0 (Pz C-3 of minor isomer), 151.1 (Pz C-3 of major isomer), 184.1 (C(O)CHCHPh). ^15^N NMR (71 MHz, CDCl_3_): *δ*_N_ ppm −161.2 (Pz N-1). HRMS (ESI^+^) for C_24_H_17_N_3_O_2_F_3_ ([M + H]^+^) calcd 436.1267, found 436.1267.

##### (2*E*)-3-(4-Chlorophenyl)-1-[1-phenyl-3-(pyridin-3-yl)-1*H*-pyrazol-4-yl]prop-2-en-1-one (**12g**)

White crystals; yield 67% (49 mg); m.p. 169.7–172.3 °C; *R_f_* = 0.50 (EtOAc/Hex 1/2, *v*/*v*). IR (*ν*_max_, cm^−1^): 3120, 3041, 1684 (C=O), 1599, 1521, 1448, 1362, 1260, 1241, 1221, 987, 977, 940, 863, 751, 705, 683. ^1^H NMR (700 MHz, CDCl_3_): *δ*_H_ ppm 6.99 (d, *J* = 15.6 Hz, 1H, C(O)CHCHPh), 7.26 (d, *J* = 8.5 Hz, 2H, CPh 3,5-H), 7.28–7.33 (m, 4H, NPh 4-H, CPh 2,6-H, Pyr 5-H), 7.42–7.45 (m, 2H, NPh 3,5-H), 7.60 (d, *J* = 15.6 Hz, 1H, C(O)CHCHPh), 7.72–7.74 (m, 2H, NPh 2,6-H), 8.08 (dt, *J*_Pyr 4-H,5-H_ = 7.9 Hz, *J* = 2.0 Hz, 1H, Pyr 4-H), 8.51 (s, 1H, Pz 5-H), 8.58 (dd, *J*_Pyr 5-H,6-H_ = 4.9 Hz, *J* = 1.6 Hz, 1H, Pyr 6-H), 8.98 (d, *J* = 2.2 Hz, 1H, Pyr 2-H). ^1^^3^C NMR (176 MHz, CDCl_3_): *δ*_C_ ppm 118.6 (NPh C-2,6), 121.9 (Pyr C-5), 122.2 (Pz C-4), 123.6 (C(O)CHCHPh), 126.9 (NPh C-4), 127.5 (Pyr C-3), 128.2 (CPh C-3,5), 128.5 (CPh C-2,6), 128.7 (NPh C-3,5), 130.4 (Pz C-5), 132.0 (CPh C-1), 135.5 (CPh C-4), 135.9 (Pyr C-4), 138.0 (NPh C-1), 141.4 (C(O)CHCHPh), 148.8 (Pyr C-6), 149.0 (Pyr C-2), 149.7 (Pz C-3), 182.8 (C(O)CHCHPh). ^15^N NMR (71 MHz, CDCl_3_): *δ*_N_ ppm −161.8 (Pz N-1), −78.2 (Pz N-1), −70.6 (Pyr N). HRMS (ESI^+^) for C_23_H_17_N_3_OCl ([M + H]^+^) calcd 386.1055, found 386.1055.

##### (2*E*)-3-(4-Methoxyphenyl)-1-[1-phenyl-3-(pyridin-3-yl)-1*H*-pyrazol-4-yl]prop-2-en-1-one (**12h**)

Yellow crystals; yield 54% (39 mg); m.p. 157–159 °C; *R_f_* = 0.51 (EtOAc/Hex 1/2, *v*/*v*). IR (*ν*_max_, cm^−1^): 3120, 3041, 1659 (C=O), 1599, 1521, 1448, 1362, 1260, 1241, 1221, 977, 940, 863, 751, 705, 683. ^1^H NMR (700 MHz, CDCl_3_): *δ*_H_ ppm 3.84 (s, 3H, OCH_3_), 6.89 (d, *J* = 8.6 Hz, 2H, CPh 3,5-H), 6.99 (d, *J* = 15.6 Hz, 1H, C(O)CHCHPh), 7.37–7.41 (m, 2H, NPh 4-H, Pyr 5-H), 7.43–7.45 (m, 2H, CPh 2,6-H), 7.51–7.54 (m, 2H, NPh 3,5-H), 7.72 (d, *J* = 15.6 Hz, 1H, C(O)CH=CHPh), 7.81–7.83 (m, 2H, NPh 2,6-H), 8.17 (d, *J*_Pyr 4-H,5-H_ = 7.9 Hz, 1H, Pyr 4-H), 8.56 (s, 1H, Pz 5-H), 8.67 (d, *J*_Pyr 5-H,6-H_ = 4.2 Hz, 1H, Pyr 6-H), 9.07 (s, 1H, Pyr 2-H). ^1^^3^C NMR (176 MHz, CDCl_3_): *δ*_C_ ppm 55.6 (OCH_3_), 114.6 (CPh C-3,5), 119.8 (NPh C-2,6), 122.1 (C(O)CHCHPh), 123.0 (Pyr C-5), 123.7 (Pz C-4), 127.4 (CPh C-1), 128.0 (NPh C-4), 128.8 (Pyr C-3), 129.9 (NPh C-3,5), 130.3 (CPh C-2,6), 131.3 (Pz C-5), 137.1 (Pyr C-4), 139.2 (NPh C-1), 144.1 (C(O)CHCHPh), 149.8 (Pyr C-6), 150.2 (Pyr C-2), 150.8 (Pz C-3), 161.9 (CPh C-4), 184.6 (C(O)CHCHPh). ^15^N NMR (71 MHz, CDCl_3_): *δ*_N_ ppm −162.2 (Pz N-1), −70.9 (Pyr N). HRMS (ESI^+^) for C_24_H_20_N_3_O_2_ ([M + H]^+^) calcd 382.1550, found 382.1550.

##### (2*E*)-3-(4-Chlorophenyl)-1-[1-phenyl-3-(pyridin-4-yl)-1*H*-pyrazol-4-yl]prop-2-en-1-one (**12i**)

White crystals; yield 55% (40 mg); m.p. 205–207 °C; *R_f_* = 0.41 (EtOAc/Hex 1/2, *v*/*v*). IR (*ν*_max_, cm^−1^): 3120, 3041, 1684 (C=O), 1599, 1521, 1448, 1362, 1260, 1241, 1221, 987, 977, 940, 863, 751, 705, 683. ^1^H NMR (700 MHz, CDCl_3_): *δ*_H_ ppm 7.07 (d, *J* = 15.6 Hz, 1H, C(O)CHCHPh), 7.34–7.37 (m, 2H, CPh 3,5-H), 7.40–7.43 (m, 3H, NPh 4-H, CPh 2,6-H), 7.52–7.55 (m, 2H, NPh 3,5-H), 7.71 (d, *J* = 15.6 Hz, 1H, C(O)CHCHPh), 7.77–7.79 (m, 2H, Pyr 3,5-H), 7.80–7.81 (m, 2H, NPh 2,6-H), 8.56 (s, 1H, Pz 5-H), 8.69–8.72 (m, 2H, Pyr 2,6-H). ^1^^3^C NMR (176 MHz, CDCl_3_): *δ*_C_ ppm 119.9 (NPh C-2,6), 123.5 (Pz C-4), 123.8 (Pyr C-3,5), 124.9 (C(O)CHCHPh), 128.2 (NPh C-4), 129.5 (CPh C-3,5), 129.7 (CPh C-2,6), 129.9 (NPh C-3,5), 131.7 (Pz C-5), 133.0 (CPh C-1), 136.8 (CPh C-4), 139.1 (NPh C-1), 140.3 (Pyr C-4), 142.9 (C(O)CHCHPh), 150.0 (Pyr C-2,6), 151.1 (Pz C-3), 184.2 (C(O)CHCHPh). ^15^N NMR (71 MHz, CDCl_3_): *δ*_N_ ppm −161.3 (Pz N-1), −70.3 (Pyr N). HRMS (ESI^+^) for C_23_H_17_N_3_OCl ([M + H]^+^) calcd 386.1055, found 386.1055.

##### (2*E*)-3-(4-Methoxyphenyl)-1-[1-phenyl-3-(pyridin-4-yl)-1*H*-pyrazol-4-yl]prop-2-en-1-one (**12j**)

Yellow crystals; yield 50% (36 mg); m.p. 167–169 °C; *R_f_* = 0.36 (EtOAc/Hex 1/2, *v*/*v*). IR (*ν*_max_, cm^−1^): 3120, 3041, 1659 (C=O), 1599, 1521, 1448, 1362, 1260, 1241, 1221, 977, 940, 863, 751, 705, 683. ^1^H NMR (700 MHz, CDCl_3_): *δ*_H_ ppm 3.84 (s, 3H, OCH_3_), 6.89–6.90 (m, 2H, CPh 3,5-H), 6.98 (d, *J* = 15.6 Hz, 1H, C(O)CHCHPh), 7.39–7.41 (m, 1H, NPh 4-H), 7.44–7.46 (m, 2H, CPh 2,6-H), 7.51–7.54 (m, 2H, NPh 3,5-H), 7.73 (d, *J* = 15.6 Hz, 1H, C(O)CHCHPh), 7.78–7.81 (m, 4H, NPh 2,6-H, Pyr 3,5-H), 8.54 (s, 1H, Pz 5-H), 8.69–8.70 (m, 2H, Pyr 2,6-H). ^1^^3^C NMR (176 MHz, CDCl_3_): *δ*_C_ ppm 55.6 (OCH_3_), 114.6 (CPh C-3,5), 119.8 (NPh C-2,6), 122.3 (C(O)CHCHPh), 123.78 (Pyr C-3,5), 123.83 (Pz C-4), 127.2 (CPh C-1), 128.1 (NPh C-4), 129.9 (NPh C-3,5), 130.4 (CPh C-2,6), 131.5 (Pz C-5), 139.2 (NPh C-1), 140.4 (Pyr C-4), 144.3 (C(O)CHCHPh), 149.9 (Pyr C-2,6), 150.9 (Pz C-3), 162.0 (CPh C-4), 184.7 (C(O)CHCHPh). ^15^N NMR (71 MHz, CDCl3): *δ*_N_ ppm −161.8 (Pz N-1), −70.4 (Pyr N). HRMS (ESI^+^) for C_24_H_20_N_3_O_2_ ([M + H]^+^) calcd 382.1550, found 382.1550.

#### 3.2.9. General Procedure for the Preparation of 3-(3-methoxy-1-phenyl-1*H*-pyrazol-4-yl)-5-phenyl-1,2-oxazole (**14**) and 5-(3-methoxy-1-phenyl-1*H*-pyrazol-4-yl)-3-phenyl-1,2-oxazole (**15**)

To a solution of *N*-hydroxy-4-toluenesulfonamide (1.56g, 8.35 mmol) in EtOH/H_2_O (9:1 *v*/*v*, (25 mL)), NaOH (0.4 g, 10 mmol) was added. A solution of appropriate chalcone **4a** or **9a** (0.3 g, 1 mmol) in EtOH (3 mL) was added to a mixture. The reaction mixture was stirred at 40 °C for 48 h; the progress of the reaction was monitored by TLC. An additional amount of NaOH (0.4 g, 10 mmol) was added and the reaction mixture was stirred at 55 °C for another 24 h. Upon completion, the reaction mixture was diluted with EtOAc (30 mL). Then, the mixture was washed with water (3 × 30 mL) and brine (30 mL). The organic layers were dried over sodium sulphate, filtrated and concentrated. The residue was purified by column chromatografy (SiO_2_, eliuent: hexane/ethyl acetate, 3/1, *v*/*v*) to produce pure **14** or **15**.

##### 3-(3-Methoxy-1-phenyl-1*H*-pyrazol-4-yl)-5-phenyl-1,2-oxazole (**14**)

Yellow solids; yield 56% (178 mg); m.p. 132–133 °C; R*_f_* = 0.66 (EtOAc/Hex 1/4, *v*/*v*). IR (*ν*_max_, cm^−1^): 3098, 2953, 1595, 1525, 1420, 1249, 1220, 1173, 950, 830, 752, 683. ^1^H NMR (700 MHz, CDCl_3_): *δ*_H_ ppm 4.16 (s, 3H, CH_3_), 6.93 (s, 1H, 4-H), 7.27–7.28 (m, 1H, NPh 4-H), 7.43–7.50 (m, 5H, NPh 3,5-H, CPh 3,5-H, CPh 4-H), 7.68–7.69 (m, 2H, NPh 2,6-H), 7.85–7.86 (m, 2H, CPh 2,6-H), 8.33 (s, 1H, Pz 5-H). ^13^C NMR (176 MHz, CDCl_3_): *δ*_C_ ppm 56.6 (CH_3_), 98.4 (Ox C-4), 98.8 (Pz C-4), 118.1 (NPh C-2,6), 125.9 (CPh C-2,6), 126.03 (Pz C-5), 126.06 (NPh C-4), 127.6 (CPh C-1), 128.9 (NPh C-3,5), 129.5 (CPh C-3,5), 130.1 (CPh C-4), 139.7 (NPh C-1), 155.4 (Ox C-3), 162.3 (Pz C-3), 169.6 (Ox C-5). ^15^N NMR (71 MHz, CDCl_3_): *δ*_N_ ppm −184.8 (Pz N-1), −120.0 (Pz N-2), −19.6 (Ox N). HRMS (ESI^+^) for C_19_H_15_N_3_NaO_2_ ([M + Na]^+^) calcd 340.1056, found 340.1061.

##### 5-(3-Methoxy-1-phenyl-1*H*-pyrazol-4-yl)-3-phenyl-1,2-oxazole (**15**)

Yellow solids; yield 37% (117 mg); m.p. 162–163 °C; R*_f_* = 0.63 (EtOAc/Hex 1/4, *v*/*v*). IR (*ν*_max_, cm^−1^): 3138, 2916, 1592, 1528, 1506, 1393, 1359, 1220, 948, 899, 752, 685. ^1^H NMR (700 MHz, CDCl_3_): *δ*_H_ ppm 4.16 (s, 3H, CH_3_), 6.77 (s, 1H, 4-H), 7.27–7.29 (m, 1H, NPh 4-H), 7.45–7.49 (m, 5H, NPh 3,5-H, CPh 3,5-H, CPh 4-H), 7.67–7.69 (m, 2H, NPh 2,6-H), 7.88–7.89 (m, 2H, CPh 2,6-H), 8.24 (s, 1H, Pz 5-H). ^13^C NMR (176 MHz, CDCl_3_): *δ*_C_ ppm 56.7 (CH_3_), 97.5 (Ox C-4), 99.0 (Pz C-4), 118.2 (NPh C-2,6), 125.6 (Pz C-5), 126.3 (NPh C-4), 126.9 (CPh C-2,6), 128.9 (CPh C-3,5), 129.3 (CPh C-1), 129.6 (NPh C-3,5), 129.9 (CPh C-4), 139.5 (NPh C-1), 161.1 (Pz C-3), 162.8 (Ox C-3), 162.9 (Ox C-5). ^15^N NMR (71 MHz, CDCl_3_): *δ*_N_ ppm −184.4 (Pz N-1), −119.7 (Pz N-2), −18.6 (Ox N). HRMS (ESI^+^) for C_19_H_15_N_3_NaO_2_ ([M + Na]^+^) calcd 340.1056, found 340.1059.

#### 3.2.10. Procedure for the Preparation of Mixture of 3-(3-methoxy-1-phenyl-1*H*-pyrazol-4-yl)-5-phenyl-^15^N-1,2-oxazole (**16**) and 5-(3-methoxy-1-phenyl-1*H*-pyrazol-4-yl)-3-phenyl-^15^N-1,2-oxazole (**17**)

To a solution of ^15^N hydroxylamine hydrochloride (139 mg, 2 mmol) and potassium hydroxide (160 mg, 4 mmol) in EtOH (96%, 15 mL), chalcone **9a** was added (304 mg, 1 mmol). The reaction mixture was stirred at 80 °C for 3 h, poured into water (30 mL) and extracted with EtOAc (3 × 50 mL). The organic layers were washed with brine (30 mL) and dried over sodium sulphate, filtrated and concentrated. The residue was purified by column chromatografy (SiO_2_, eliuent: hexane/ethyl acetate, 3/1, *v*/*v*) to produce an inseparable mixture of compounds **16** and **17** with a 23% yield. The inseparable mixture of regioisomers **16** (major) and **17** (minor) was obtained in a ratio of about 8:1.

Yellow solid; yield 23% (73 mg mixture); ^1^H NMR (700 MHz, CDCl_3_): *δ*_H_ ppm 4.15 (s, 3H, CH_3_ of minor regioisomer), 4.16 (s, 3H, CH_3_ of major regioisomer), 6.77 (d, *J* = 1.23 Hz, 1H, Ox 4-H of major regioisomer), 6.93 (d, *J* = 1.31 Hz, 1H, Ox 4-H of minor regioisomer), 7.27–7.29 (m, 1H, NPh 4-H of both regioisomers), 7.45–7.49 (m, 5H, NPh 3,5-H, CPh 3,5-H, CPh 4-H of both regioisomers), 7.67–7.69 (m, 2H, NPh 2,6-H of both regioisomers), 7.84–7.86 (m, 1H, NPh 2,6-H of minor regioisomer), 7.87–7.89 (m, 2H, CPh 2,6-H of major regioisomer), 8.24 (s, 1H, Pz 5-H of major regioisomer), 8.33 (s, 1H, Pz 5-H of minor regioisomer). ^13^C NMR (176 MHz, CDCl_3_): *δ*_C_ ppm 56.62 (CH_3_ of minor regioisomer), 56.73 (CH_3_ of major regioisomer), 97.51 (d, ^2^*J_C,N_* = 1.23 Hz, Ox C-4 of major regioisomer), 98.45 (d, ^2^*J_C,N_* = 1.11 Hz, Ox C-4 of minor regioisomer), 98.82 (d, ^2^*J_C,N_* = 8.40 Hz, Pz C-4 of minor regioisomer), 98.96 (d, ^3^*J_C,N_* = 0.66 Hz, Pz C-4 of major regioisomer), 118.12 (NPh C-2,6 of minor regioisomer), 118.20 (NPh C-2,6 of major regioisomer), 125.53 (Pz C-5 of major regioisomer), 125.88 (CPh C-2,6 of minor regioisomer), 126.03 (d, ^3^*J_C,N_* = 1.47 Hz, Pz C-5), 126.06 (NPh C-4 of minor regioisomer), 126.27 (NPh C-4 of major regioisomer), 126.91 (d, ^3^*J_C,N_* = 2.28 Hz, CPh C-2,6 of major regioisomer), 127.59 (CPh C-1 of minor regioisomer), 128.85 (CPh C-3,5 of major regioisomer), 128.94 (NPh C-3,5 of minor regioisomer), 129.28 (d, ^2^*J_C,N_* = 7.11 Hz, CPh C-1 of major regioisomer), 129.50 (CPh C-3,5 of minor regioisomer), 129.54 (NPh C-3,5 of major regioisomer), 129.89 (CPh C-4 of major regioisomer), 130.06 (CPh C-4 of minor regioisomer), 139.51 (NPh C-1 of major regioisomer), 139.68 (NPh C-1 of minor regioisomer), 155.35 (d, ^1^*J_C,N_* = 2.25 Hz, Ox C-3 of minor regioisomer), 161.09 (Pz C-3 of major regioisomer), 162.27 (d, ^3^*J_C,N_* = 1.92 Hz, Pz C-3 of minor regioisomer), 162.77 (d, ^1^*J_C,N_* = 2.89 Hz, Ox C-3 of major regioisomer), 162.84 (d, ^2^*J_C,N_* = 1.39 Hz, Ox C-5 of major regioisomer), 169.61 (d, ^2^*J_C,N_* = 1.52 Hz, Ox C-5 of minor regioisomer). ^15^N NMR (71 MHz, CDCl_3_): *δ*_N_ ppm −184.8 (Pz N-1 of minor regioisomer), −184.5 (Pz N-1 of major regioisomer), −119.9 (Pz N-2 of minor regioisomer), −119.7 (Pz N-2 of major regioisomer), −19.5 (Ox N of minor regioisomer), −18.6 (Ox N of major regioisomer). HRMS (ESI^+^) for C_19_H_15_^15^NN_2_NaO_2_ ([M + Na]^+^) calcd 340.1056, found 340.1061.

## 4. Conclusions

In conclusion, we have synthesized novel diverse pyrazole-chalcone derivatives, starting with easily accessible 3-hydroxy-1-phenyl-1*H*-pyrazole via the Claisen–Schmidt condensation reaction of intermediate 4-acetyl or formyl-1-phenyl-1*H*-pyrazoles. The condensation of 4-formyl-1-phenyl-1*H*-pyrazoles with various acetophenones, as well as the reaction of 4-acetyl-3-hydroxy-1-phenyl-1*H*-pyrazoles with different carbaldehydes, produced appropriate (*E*)-chalcones. The reaction of 4-acetyl-3-pyridinyl-phenyl-1*H*-pyrazoles with acetophenones caused the formation of mixtures of compounds with predominant (*E*)-chalcones and less stable (*Z*)-chalcones in different ratios. Extensive NMR spectroscopic studies have been undertaken using standard and advanced methods to unambiguously determine the structure and configuration of novel, pyrazole-chalcone regioisomeric 3(5)-(1*H*-pyrazol-4-yl)-5(3)-phenyl-1,2-oxazole derivatives.

## Data Availability

The data that support the findings of this study are available upon request.

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
