# Peer review of "Synthesis and Characterization of Novel Heterocyclic Chalcones from 1-Phenyl-1H-pyrazol-3-ol"

_molecules, 2022, doi:10.3390/molecules27123752_

Round 1

Reviewer 1 Report

The article "Synthesis and Characterization of Novel Heterocyclic Chalcones from 1-Phenyl-1H-pyrazol-3-ol" by Urbonavičius et al. presents the synthesis and characterization of new heterocyclic chalcones with the structure of 1-phenyl-1H-pyrazol-3-ol. All research planned and carried out in a logical way.
Correctly written introduction perfectly introduces the subject of the article. Factual and specific. The discussion of the results was carried out very well. Particularly noteworthy is the graphic design of the entire article, which is really a pleasure to read. I am impressed with the discussion of NMR spectroscopy of the compounds obtained. The authors have demonstrated proficiency and extensive experience here. Congratulations. The part on Materials and Methods was written without any major comments affecting the substantive reception of the whole. The presented recipes allow you to easily repeat the research. The summary resulting from the discussion and presented results highlights the most important results presented in the article.

Minor remarks:
1) Scheme 1, p. 3 and Scheme 3, p. 6: please give R1 and R2 for two R substituents in one formula.
2) Scheme 3, p. 6: "in accordance with ref [56]." - please enter the conditions, which will facilitate the receipt of the entire schematic (ref [56] can of course be left)
3) Lines 219-222, p. 8: "The NMR spectra of inseparable mixtures showed the presence of both isomers in different ratios with a predominance of the E-isomer. In contrast, compounds 12g – j were obtained only as pure E-isomers. " - please explain why.
4) What EtOH was used by the authors in their syntheses? Please clarify.
5) In section 3.1. please add manufacturers, cities and countries of equipment and materials used where they are missing.
6) For the rest of the article, please change "ml" to "mL".

Reviewer 2 Report

The paper describes preparation of chalcones bearing pyrazole derived moiety in 1- or 3-position, and further transformation of C=CC=O fragment into respective 1,2-oxazole. Although the chemistry is simple and obvious, the authors developed effective  procedures for a variety of biorelevant compounds and established their structures and configuration of C=C bond with the use of  advanced NMR spectroscopy techniques and isotopically labeled compounds. Presented in paper detailed NMR characteristics and subtle features of the structure for the prepared compounds are interesting in themselves and can be used in structural studies of related systems.

At the same time, there are some points that should not be ignored:

- p.2, lines 51-52. Authors claimed: Most chalcones are formed and exist as thermodynamically more stable E-isomers, as the Z-conformers are unstable due to steric effects between the carbonyl group and one of the rings.

1) Can Z-isomers be called thermodynamically unstable? They are thermodynamically less stable than E-isomers but not obviously “unstable”. Their stability depends on activation energy and they may be quite stable at ordinary conditions.

2) It is not clear why isomers with E-configuration are called by authors “E-isomers”, and with Z-configuration, ”Z-conformers”?? Conformers correspond to different arrangements of atom in a molecule arising from rotation about single bond.

-p.9, line 262: “The regioselective formation of 1,2-oxazoles 14 or 15 can be assigned to the tosyl moiety of TsNHOH which contributes to the greater nucleophilicity of a nitrogen atom [61].

It is rather surprising that electron accepting tosyl group “contributes to the greater nucleophilicity of a nitrogen atom”. Strictly speaking, the sources cited ([61]) do not provide any data confirming the above. The situation is more complicated. The tosyl group affects the acidity of the NH group and the relative acidity of the NH and OH groups, as well as the ability to form anions.

Strictly speaking, in the presented references ([61) no data were presented to confirm the above statement.  The situation is more complex. The tosyl group influences the acidity of NH group and relative acidity of NH and OH groups and  the ability for anion generation. TsNHOH acts simultaneously as N-H acid and N-nucleophile; because of this TsNHOH catalyzes acetal formation (JOC, 1970, 35, 1962). Quantitative data relevant to this issue are discussed in OrgBiomolChem 2003 1176-1180.

Minor technical shortcomings:

p.5, line 153: “rt, and 24 h”. Please, remove “and” in this caption and captions to other Schemes

p.7, line 198: correct “pirid” for pyrid. Please, correct throughout manuscript.

Identified places that require correction are highlighted in yellow in the attached file.

I recommend paper for publication after considering above comments.

Reviewer 3 Report

The manuscript by Urbonavicius, A. et al. describes a synthetic route to construct diverse pyrazole-based chalcones. Based on the aim of the journal, which considers all original research manuscripts provided that the work reports scientifically sound experiments and provides a substantial amount of new information. I will provide my comments, in general, it is a very complete work, it is a comprehensive and well-described research that could be published once the items listed below have been addressed:

1)    The authors have developed a complete methodology that has been used to generate a library of compounds published in different articles. As it is research focused on obtaining new molecules, considering the heterocycle (pyrazole-chalcones) has proved to be part of studies with biological tests. There are different works to be considered.

Nidhar, M., Sonker, P., Sharma, V.P. et al. Design, synthesis and in-silico & in vitro enzymatic inhibition assays of pyrazole-chalcone derivatives as dual inhibitors of α-amylase & DPP-4 enzyme. Chem. Pap. 76, 1707–1720 (2022). https://doi.org/10.1007/s11696-021-01985-1

Macarini, A.F., Sobrinho, T.U.C., Rizzi, G.W. et al. Pyrazole–chalcone derivatives as selective COX-2 inhibitors: design, virtual screening, and in vitro analysis. Med Chem Res 28, 1235–1245 (2019). https://doi.org/10.1007/s00044-019-02368-8

Jadhav, S.Y., Peerzade, N.A., Hublikar, M.G. et al. Synthesis and Pharmacological Screening of Difluorophenyl Pyrazole Chalcone Conjugates as Antifungal, Anti-Inflammatory, and Antioxidant Agents. Russ J Bioorg Chem 46, 1128–1135 (2020). https://doi.org/10.1134/S1068162020060102

Rupireddy, V., Chittireddy, V.R.R. & Dongamanti, A. An Efficient Approach for the Synthesis of Triazole Conjugated Pyrazole Chalcone Derivatives. Chemistry Africa 3, 45–52 (2020). https://doi.org/10.1007/s42250-019-00103-9

Therefore, you should indicate an analysis of why the yields are very different (2, 3, 4, 4, 5 and their derivatives), what is the problem, conversion, or purification, are there other stereoisomers? Support your answer with NMR, GCMS and HPLC data.

2)    Proofread document and take care of typing errors, for example: pag. 6, line 169, exellent.

3)    Explain, why 5b in the deprotection process, the CH3O group is not deprotected, as well as low yields.? NMR, HPLC or CGMS data must be added.

4)    The yield of the Fries rearrangement is not reported.

5)    Although some of the methodologies employed have been reported, the authors did not make any adjustments and/or optimizations. In the case of Claisen-Schmidt products, the crude of the two series should be placed in the SI. Also, an explanation for this behavior should be given.

6)    The following sentence is indicated “The formation of intermediate reaction products such as isoxazolines or oximes could be also identified by LC/MS and NMR data” but no LC/MS data is observed in the SI. It is important to add the reaction crude.

7)    Please, High-resolution mass spectra must be shown for all compounds.

Reviewer 4 Report

The manuscript entitled “Synthesis and Characterization of Novel Heterocyclic Chalcones from 1-Phenyl-1H-pyrazol-3-ol” by Sackus and co-workers, describes the preparation and characterization of a new family of chalcone derivatives containing a pyrazole ring. Although the novelty of the work is limited, the reported methodology provides good yields of a wide number of new compounds that are characterized in great detail. Further transformation of the pyrazole containing chalcones to oxazoles is also reported, including the preparation of 15N labelled 1,2-oxazoles.

The manuscript is very well written, including citations of relevant precedents and related works. The methods section is written with detail although some minor points should be addressed to ensure reproducibility of the results (see below). Identity of the compounds is clearly demonstrated and the conclusions are consistent with the results reported. Figures should be improved to help better understand the results reported (see below).

Therefore, I recommend publication of the paper after minor revision, covering the points listed below:

  • Scheme 1 is splitted into two pages, please redraw it as a single body Scheme.
  • Revise Figures 2, 3, 4 and Schemes 5 and 6, to include all labels referred to in the text in the corresponding structures (i.e. C-4 is indicated in the text referring to Figure 2 but the label does not appear in the structure, and the same happens with C-2’’’ and C-4’’’ for Figure 4, C-3’ and C-5’ for Scheme 6).
  • In line 187 the text refers to an olefinic carbon resonating at 142.7 whereas in Figure 3 the chemical shift value depicted is 142.6. Please correct.
  • Scheme 5 needs to be redrawn to make it more clear. As it appears now, the reaction does not seem to be regioselective. Please slip it into three equations, showing the selective reactions of 4a and 9a with TsNHOH and the unselective reaction of 9a with hydroxylamine. Please increase the size of the resulting Scheme.
  • A note should be added at the “Materials and Methods” section to define the labels used on describing the NMR data.
  • Use of NaH in DMF has been reported to have explosion hazards associated ( Process Res. Dev.2019, 23, 10, 2210–2217). Please make reference to it.
  • Line 381, two carbon signals (127.4 and 139.3) are assigned to the same carbon (C-5) in 3b. The same problem is found in line 390 for compound 3c.
  • The yield reported for compound 4a in Scheme 1 (61%) does not match with the one reported in the “Materials and Methods” section (58% line 402). The same applies to 4g (66% in Scheme 1 and 65% in line 480).
  • Two carbon signals at 128.6 and 128.7 in the 13C NMR description for compound 4g are not assigned (line 488). The same applies to 4i (carbon signals at 136.9 and 137.0, line 518.
  • The yield reported for compound 8b in Scheme 3 (70%) does not match with the one reported in the “Materials and Methods” section (53% line 608).
  • Reaction time for products 8a-l shown in Scheme 3 should be indicated (either in the Scheme or in the “Materials and Methods” section). The same applies to compounds 9.
  • Line 600, the integration in the signal at 7.69 (1H) is missing.
  • Several assignments are missing for compounds 8: 8b (signals between 7.51 and 7.86 in the 1H NMR description), 8c (signals between 7.80 and 7.84 in the 1H NMR spectra and signals between 127.1 and 135.3 in the 13C NMR spectra), 8e (signals at 7.69 and 7.74 in the 1H NMR description, and several carbons in the 13C NMR), just to cite a few. Please make predictive assignments or a general sentence stating the difficulties on its assignment.
  • Line 847-848: please indicate the complete reaction time for the reaction (it seems that MW heating is carried out in 10 minutes intervals and controlled by TLC but the total reaction time is not noted).
  • In all of the reactions to obtain compounds 12b-j/13b-j, the amount of product isolated is 50 mg? Please revise.

Round 2

Reviewer 3 Report

To the authors, thank you for answering my questions, I am sure that your work is excellent.